# Resolving the spatial architecture of myeloma and its microenvironment at the single-cell level

Lukas John [1,2,22], Alexandra M. Poos[1,2,22], Alexander Brobeil[3], Carolina Schinke [4], Stefanie Huhn[1], Nina Prokoph[1,2], Raphael Lutz[1,5,6], Barbara Wagner[1], Maurizio Zangari[4], Stephan M. Tirier[7], Jan-Philipp Mallm[8], Sabrina Schumacher [7], Dominik Vonficht[5,6], Llorenç Solé-Boldo[9,10,11], Sabine Quick[1], Simon Steiger [7], Moritz J. Przybilla[12,13], Katharina Bauer[8], Anja Baumann[1,2], Stefan Hemmer[14], Christoph Rehnitz[15], Christian Lückerath[15], Christos Sachpekidis[16,17], Gunhild Mechtersheimer[3], Uwe Haberkorn[16], Antonia Dimitrakopoulou-Strauss[16,17], Philipp Reichert[1], Bart Barlogie[4], Carsten Müller-Tidow [1,18], Hartmut Goldschmidt [1,18], Jens Hillengass [19], Leo Rasche [4,20,21], Simon F. Haas[5,9,10,11], Frits van Rhee [4], Karsten Rippe [7], Marc S. Raab [1,2], Sandra Sauer[1,23] & Niels Weinhold [1,2,4,23] ✉

In multiple myeloma spatial differences in the subclonal architecture, molecular signatures and composition of the microenvironment remain poorly characterized. To address this shortcoming, we perform multi-region sequencing on paired random bone marrow and focal lesion samples from 17 newly diagnosed patients. Using single-cell RNA- and ATAC-seq we find a median of 6 tumor subclones per patient and unique subclones in focal lesions. Genetically identical subclones display different levels of spatial transcriptional plasticity, including nearly identical profiles and pronounced heterogeneity at different sites, which can include differential expression of immunotherapy targets, such as CD20 and CD38. Macrophages are significantly depleted in the microenvironment of focal lesions. We observe proportional changes in the T-cell repertoire but no site-specific expansion of T-cell clones in intramedullary lesions. In conclusion, our results demonstrate the relevance of considering spatial heterogeneity in multiple myeloma with potential implications for models of cell-cell interactions and disease progression.

Intra-tumor heterogeneity is a hallmark of the plasma cell malignancy multiple myeloma (MM)[1,2]. In order to decipher this heterogeneity, the genomic landscape of MM has been extensively studied using bulk sequencing techniques, including whole exome and whole genome sequencing (WGS)[3–6]. Recently, high throughput single-cell (sc) RNA sequencing (scRNA-seq) was successfully applied to describe the intra-tumor heterogeneity in newly diagnosed and relapsed-refractory MM patients, the changes in the tumor microenvironment (TME) associated with disease progression and the subclonal evolution during treatment[7–10]. These studies have significantly contributed to our understanding of MM biology by dissecting intra-tumor heterogeneity and TME interactions within a randomly selected site at the iliac crest.

However, intra-tumor heterogeneity in MM also arises from focal lesions, which are nodular accumulations of malignant plasma cells scattered throughout the bone marrow (BM). The number and size of focal lesions are associated with poor prognosis[11,12]. Furthermore, a recent multi-region sequencing study demonstrated distinct driver mutations in them[5]. Thus, the critical step toward disease progression could be the tumor evolution in spatially restricted areas in the skeletal system that would not necessarily be apparent from the analysis of a single biopsy from the iliac crest.

In this work we examine this hypothesis by conducting a prospective imaging-guided sampling of BM from focal lesions. By subjecting this unique biorepository to a comprehensive bulk- and sc-sequencing analysis we prove the concept of MM as a spatially heterogeneous disease, show gene signatures associated with focal lesions and demonstrate spatial heterogeneity in the TME at the single-cell level.

## Results

### Spatial tumor heterogeneity is a frequent event in newly diagnosed myeloma patients

We started the analysis by determining the extent of spatial heterogeneity in our cohort of newly diagnosed MM patients using WGS (Fig. 1a, b). Our sample set comprised bone marrow from a random iliac crest site (RBM) and paired focal lesion specimens, including 11 intra- and five paramedullary (=soft-tissue component arising from bone) lesions. In most patients ($n = 12/16$) we found major differences in chromosomal and/or mutational profiles, defined by the presence of unshared or enriched events (Fig. 1c, Supplementary Data 1). In all patients with major differences, the dominant subclone at the focal lesion site was not detectable or just a minor subclone in the RBM (Fig. 1d, Supplementary Fig. 1). Compared to intramedullary lesions, there was a trend to a higher proportion of heterogeneous (=unshared + enriched) mutations in patients with paramedullary disease (mean 33.4% (range: 9.65–58.83%) vs. 14.6% (1.02–41.08%), $p = 0.07$, two-sided Wilcoxon rank sum test. Supplementary Fig. 2a). The proportion of heterogeneous mutations was not associated with the high-risk aberration gain(1q) neither in our data nor in a recently published multi-region sequencing study[5] ($p > 0.05$, Supplementary Fig. 2b, c).

Site-unique or strongly enriched non-synonymous single nucleotide variants (SNVs) affecting MM driver genes, such as *KRAS*, *NRAS* and *TP53*, were seen in 6 patients (Fig. 1d, e, Supplementary Fig. 1, Supplementary Table 1). For three of these patients scRNA-seq was available with RAS-mutations being accessible. Importantly, due to the sparse nature of scRNA-seq measurements, MM cells with a wildtype call presumably contained undetected RAS-mutations. Patient P02 presented with two *KRAS* and one *NRAS* mutation and according to WGS, *KRAS*[G38A] (p.G13D) was unique to the focal lesion (Fig. 1e). Using scRNA-seq, we observed the mutation in 651/1442 focal lesion cells and in only 3/814 RBM cells with detectable *KRAS* expression, suggesting that it was present at both BM sites but indeed strongly enriched in the focal lesion (Fig. 1e). The other two *RAS* mutations in this patient, *KRAS*[G35A] (p.G12D) and *NRAS*[G38A] (p.G13D), were enriched in the RBM (Fig. 1e, Supplementary Fig. 2d–f). In patients P01 and P04, enrichment of *KRAS*[G35T] (p.G12V) and *KRAS*[A183C] (p.Q61G) at one BM site could be seen, respectively, further supporting WGS data (Supplementary Fig. 2g–j).

Taken together, we frequently observed spatial tumor heterogeneity in MM patients with focal lesions, including site-enriched SNVs affecting known myeloma driver genes.

### Site-unique subclones are present in focal lesions

Having shown the presence of unique or strongly enriched mutations in focal lesions, we next sought to define the spatial subclonal architecture in more detail using sc-sequencing. ScRNA-seq data was available for 5 focal lesion/RBM pairs, and sc Assay for Transposase-Accessible Chromatin using sequencing (scATAC-seq) data could be generated for 4 of them (Fig. 1b, Supplementary Fig. 3). We defined subclones based on the presence of subclonal copy number aberrations (CNAs). For scRNA-seq, CNAs were inferred using InferCNV[13], while for scATAC-seq we applied an approach published by Lareau and co-workers[14]. To improve the accuracy of CNA-calls, we used WGS to supervise the analysis (please see methods for more details). Patient P05 with two prominent and two rare subclones in scRNA-seq and scATAC-seq is shown as an example in Fig. 2a. Usually there was high concordance between the two sc-sequencing methods (Supplementary Fig. 4). Yet, in P04 scRNA-seq indicated a subclone with deletion of chromosome 14 but no deletion of chromosome 13 (subclone 6B in Supplementary Fig. 4d), which could not be confirmed using WGS and scATAC-seq. This highlights the value of bulk WGS as a confirmatory method if CNA-calls from sc-techniques are used for detection of subclones. Considering only subclones with CNAs, which were confirmed by WGS or the second single-cell method, we found a median of 6 subclones per patient (range: 4–8) (Fig. 2b, c). This number is higher compared to a recent scRNA-seq study, where up to 3 transcriptional subclones were seen[9] but comparable to another sc study based on SNVs, which observed up to 6 subclones[15]. Comparing paired samples we detected site-unique CNA-subclones in three patients (Fig. 2b). While in patient P01 each BM site presented with a dominant unique subclone, up to two unique subclones per site were seen in P02 and P04 (Fig. 2c).

In summary, using a supervised CNA-analysis we observed a median of 6 subclones per patient, including site-unique subclones, illustrating that clonal heterogeneity could be underestimated when including only a single BM specimen.

### Expression signatures associated with focal lesions

We next addressed the question if the solid-tumor-like growth pattern, which characterizes focal lesions, is associated with a unique gene expression profile. We used bulk RNA-seq to compare paired CD138-enriched samples ($n = 11$ pairs) and applying the Wald test in DESeq2[16], we found 47 transcripts, which were differentially expressed after Benjamini-Hochberg correction for multiple testing (Fig. 3a, Supplementary Data 2). To validate this finding, we analyzed microarray gene-expression profiling (GEP) data of paired focal lesion/RBM samples from the University of Arkansas for Medical Sciences (UAMS) ($n = 250$ pairs)[5]. Using Wilcoxon signed rank tests for paired samples, we confirmed differential expression for 6 of the transcripts, including up-regulation of *MYLIP* and *ADM* as well as down-regulation of the two chemokines *CXCL7 (PPBP)* and *CXCL12* in focal lesions (Fig. 3a, b, Supplementary Data 2). Of note, 28/47 transcripts, including several non-coding genes, could not be assessed using GEP, which could explain the rather low number of confirmed differences. Differential expression of *MYLIP*, *ADM*, *CXCL7*, and *CXCL12* was seen in both intra- and paramedullary lesions, but the downregulation of *CXCL12* was slightly more pronounced in paramedullary disease (mean log2-fold change paramedullary: 4.16 (range = 1.01–7.56); intramedullary: 1.63 (range = 0.76–2.89), $p = 0.32$ in two-sided Wilcoxon rank sum test, Supplementary Fig. 5a). There were no CNAs that could explain the consistent differential expression of the four genes (Supplementary Data 1), and the same holds true for the proportion of Ki67-positive MM cells in core biopsies, which was not significantly different between paired focal lesion and RBM ($n = 8$ pairs, focal lesion median: 5.7% (range: 0.2–48.6%) vs. RBM median: 9.2% (2.7–16.7%), $p = 0.84$ in two-sided Wilcoxon signed rank test, Fig. 3c). In order to understand the clinical implications of these differentially expressed genes, we correlated their expression with patient characteristics and outcome in a cohort of 653 newly diagnosed MM patients. Low expression (≤median) of *CXCL7* (hazard ratio (HR): 1.39, 95% confidence interval: 1.26–1.52, $p = 0.01$, Cox regression and log-rank test) and *CXCL12* (HR: 1.71

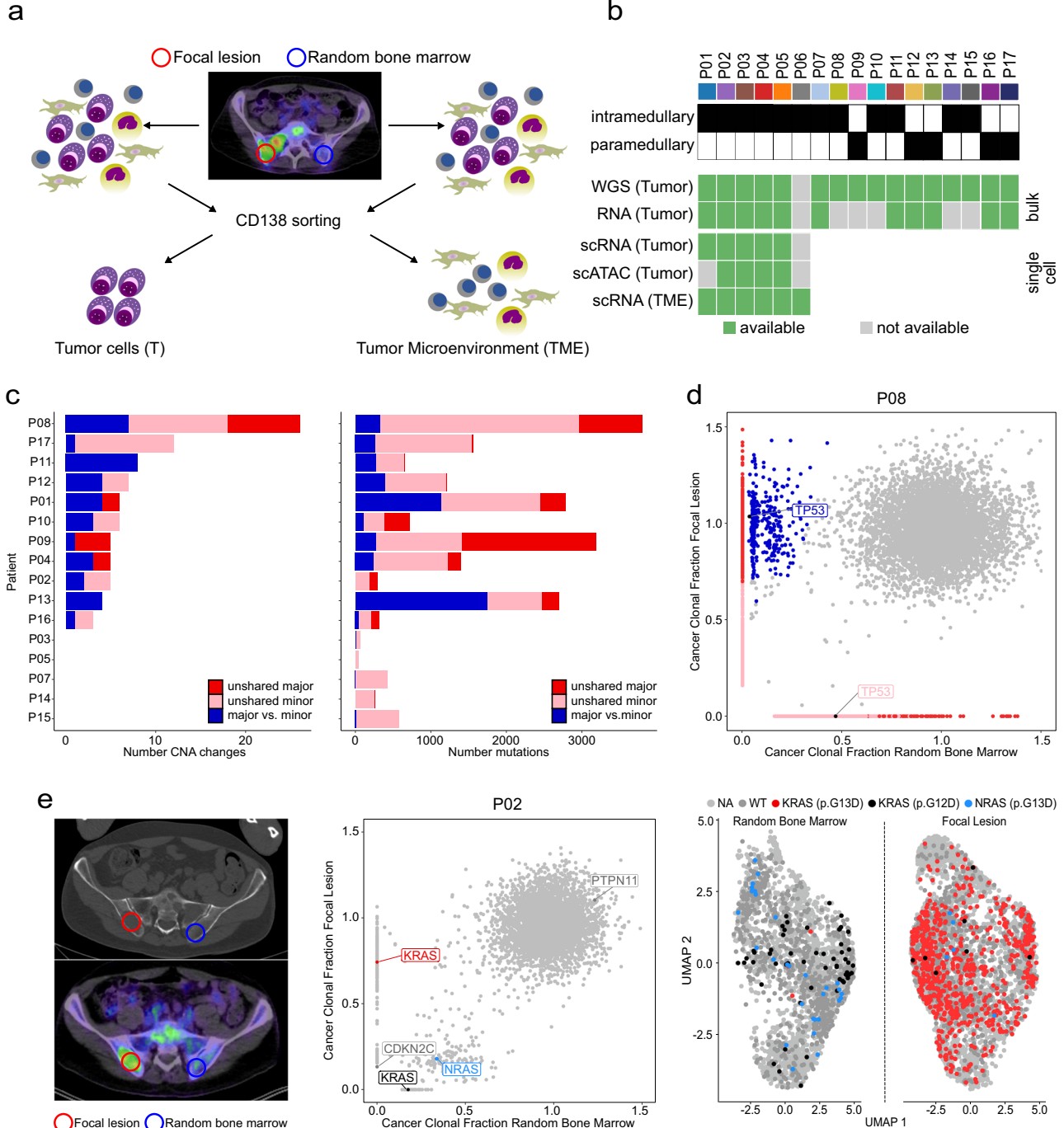

**Fig. 1 | Sample origin, applied methods and the extent of spatial heterogeneity.**
**a** Random bone marrow (RBM) aspirates from the iliac crest and imaging-guided samples from focal lesions (FL) were processed using CD138-positive selection. **b** CD138-positive tumor and CD138-negative tumor microenvironment (TME) samples from intra- or paramedullary components were analyzed using bulk and single-cell sequencing. *Gray* squares: samples not available. For P06, only TME data was available. **c** Number of copy number aberrations (CNA, >200 kb) and mutational differences between paired RBM and FL samples. Left panel: Red and pink denote major and minor *unshared* CNAs, respectively. *Blue* denotes CNAs that dominated in one sample (cancer clonal fraction (CCF) > 0.6) but were only a minor subclone in the paired sample. Right panel: Number of major (red), and minor (pink) unshared single-nucleotide variants (SNVs). Major SNVs with a

threefold enrichment between the paired samples were classified as enriched (blue). **d** Whole genome sequencing CCF plot for total SNVs in paired RBM/FL specimens from patient P08 as an example for a patient with two site-unique, biallelic *TP53*-mutations. The color code corresponds to the one in (**c**). **e** Imaging, whole genome and single-cell RNA-seq data for patient P02. Left panel: CT- and PET-CT-scans showing the location of the sampled FL (red circle) and RBM (blue circle). Middle panel: CCF-plot for SNVs in paired specimens. The three unique *RAS* mutations are depicted in *red* (KRAS p.G13D), *blue* (NRAS p.G13D) and *black* (KRAS p.G12D), respectively. Uniform Manifold Approximation and Projection (UMAP) and single-cell calls for these three SNVs are shown in the right panel. Dark gray dots denote cells with a wild type (WT) call, light gray dots indicate cells with no call. Source data are provided as a Source Data file.

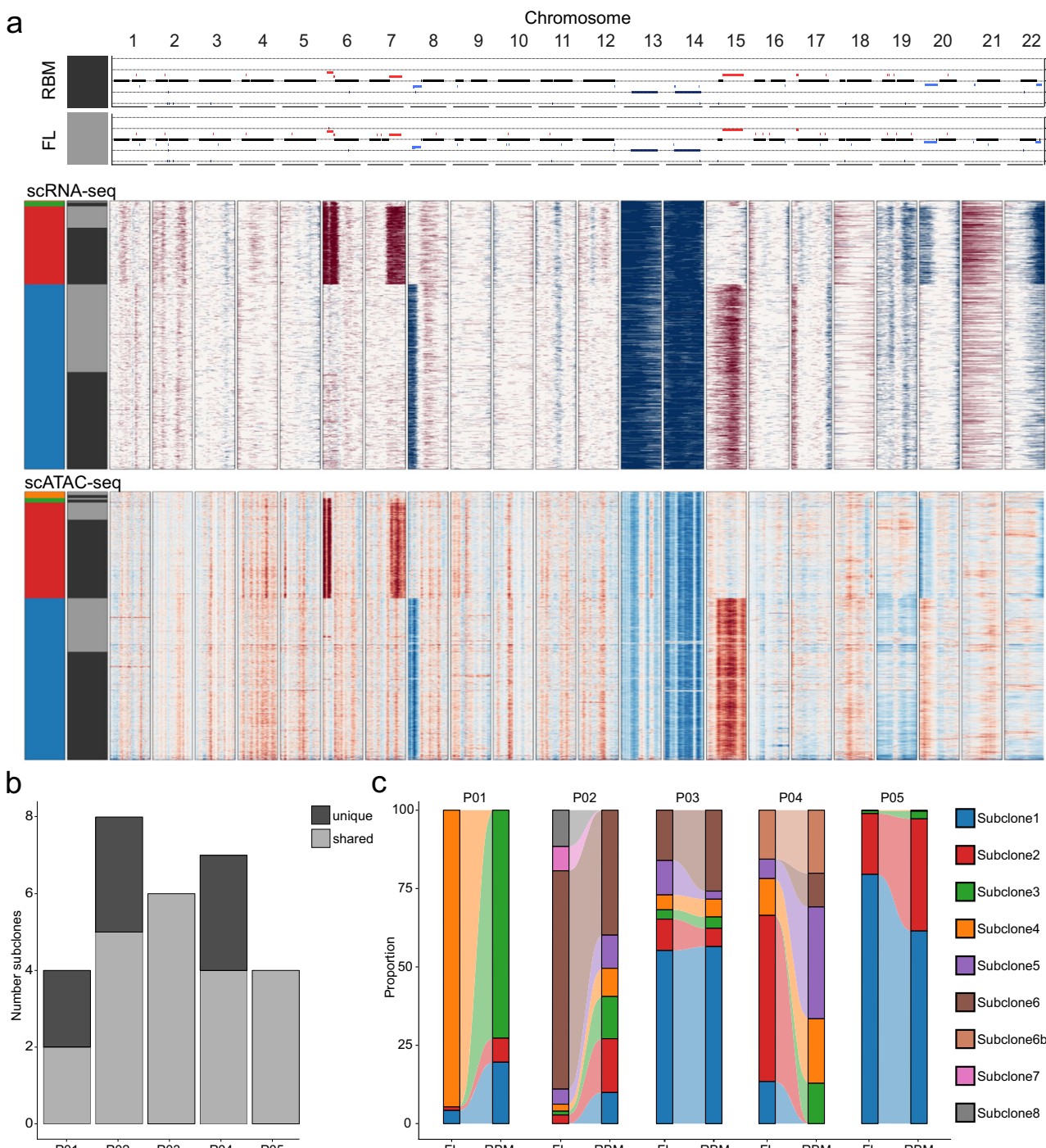

**Fig. 2 | The spatial subclonal architecture. a** Data for patient P05 is shown as an example for our approach for detection of subclones, which is based on subclonal copy number aberrations (CNAs). In the upper panel the whole genome sequencing chromosomal profiles for each autosomal chromosome in paired focal lesion (FL, gray bar) and RBM (*black* bar) are depicted. *Light red* and *blue* denote subclonal chromosomal gains and losses, respectively. Clonal events are marked with dark red and blue. To identify subclones, the average relative gene expression/accessibility in regions impacted by subclonal events was used for supervised clustering of scRNA-seq (middle panel) and scATAC-seq (lower panel) data of paired FL (gray bar) and RBM (black bar) samples. In the heatmaps *red* and *blue* signals correspond to higher and lower gene expression/accessibility, respectively. The four detected subclones are depicted on the left side of the two heatmaps in blue, red, green and orange, respectively. **b** Number of detectable CNA-defined subclones for each patient. For 3 patients (P01, P02 and P04), unique subclones, which were only detectable at one bone marrow site, are shown in dark color. **c** Proportion of CNA-defined subclones in paired FL/RBM scRNA-seq data for each patient. Subclone 6b in P04 corresponds to tumor cells with a deletion of chromosome 14 but no deletion of chromosome 13, which could not be seen in scATAC-seq and WGS (please also see Supplementary Fig. 4d). Source data are provided as a Source Data file.

(1.58–1.84), $p < 0.001$, Cox regression and log-rank test) was associated with inferior overall survival (Supplementary Fig. 5b, c). No significant effect was seen for *ADM* and *MYLIP* (Supplementary Fig. 5d, e). For *CXCL7* and *CXCL12*, low expression was associated with increased

plasma cell (PC) infiltration levels and advanced disease according to the revised International Staging System (rISS) ($p < 0.05$, t-test and chi-square test, respectively, Supplementary Data 3). Furthermore, low expression of *CXCL12* was associated with cytogenetic risk markers,

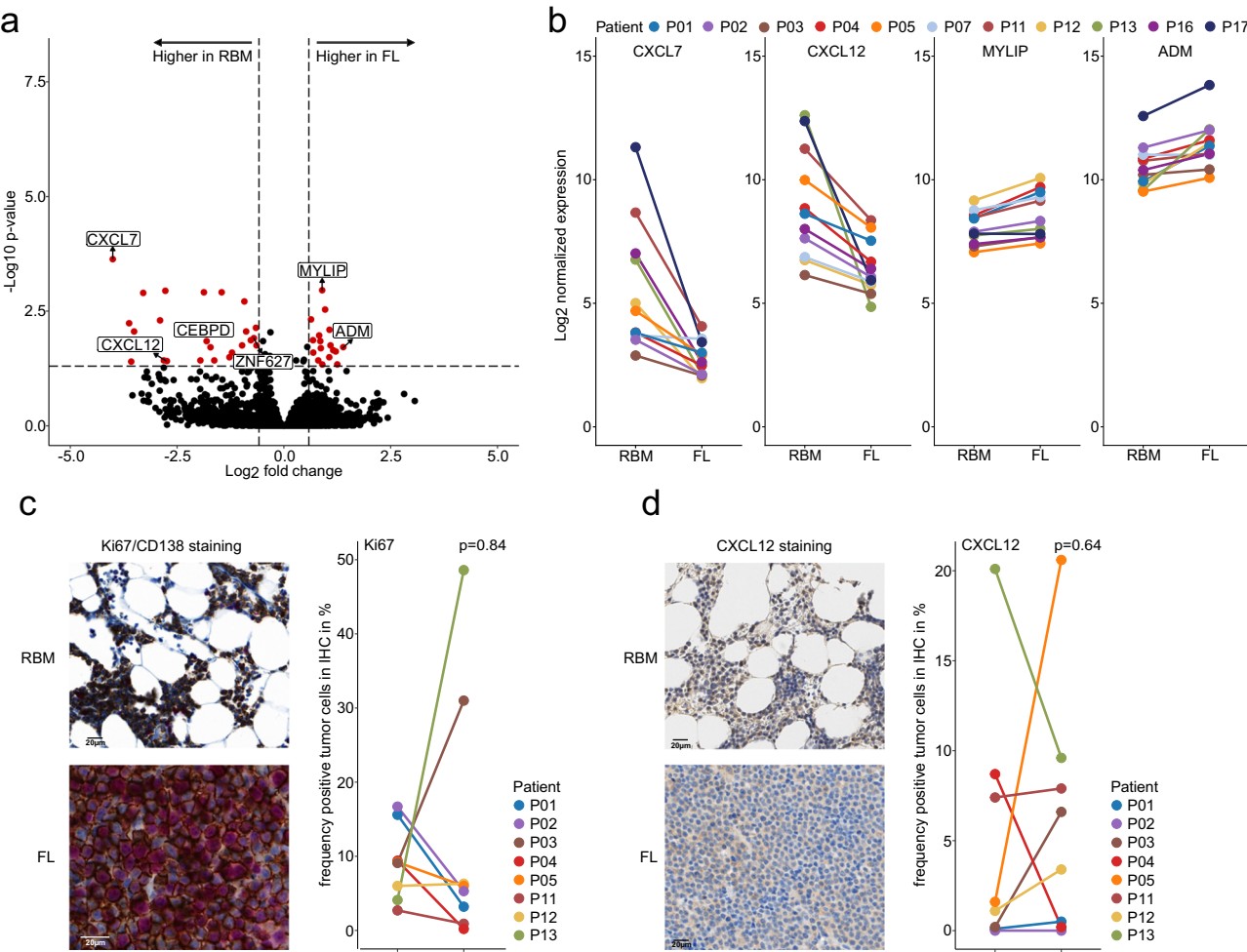

**Fig. 3 | Differentially expressed genes between paired samples. a** Volcano plot for the comparison of paired focal lesion (FL)/random bone marrow (RBM) bulk RNA-seq from 11 patients using the two-sided Wald test. The 47 genes, which showed a ≥1.5-fold difference between focal lesion and RBM, and which were significant ($p < 0.05$) in Wald-test after Benjamini-Hochberg correction for multiple testing, are depicted in *red*. The 6 genes, for which differential expression ($p < 0.05$) could be confirmed using microarray data for 250 focal lesion/RBM pairs from the University of Arkansas for Medical Sciences, are highlighted. **b** Line plots for the log2-normalized bulk RNA-seq expression values of the differentially expressed genes *ADM*, *CXCL7 (PPBP)*, *CXCL12* and *MYLIP* are shown. **c** Ki-67 index of myeloma cells. Slides were co-stained for CD138 (DAB) and Ki-67 (FastRed). Scale bar indicates 20 μm. The RBM and the FL of patient P13 are shown in the left panel. In the right panel a line plot is shown for the proportion of Ki67-positive plasma cells in paired samples from 8 patients. **d** Proportion of CXCL12-positive myeloma cells. In the left panel paired tissue slides from patient P13, which were stained for CXCL12, are depicted as examples. Since MUM1/CXCL12 double stainings were not feasible, plasma cells were identified based on their morphology and location in consecutive sections stained with MUM1. Scale bar indicates 20 μm. The line plot in the right panel shows the proportion of CXCL12-positive myeloma cells in paired samples from 8 patients. The *p* values in (**c**, **d**) were calculated using the two-sided Wilcoxon signed-rank-test. Due to limited patient material, there are no independent replicates for immunohistochemistry. Source data are provided as a Source Data file. Images are representative for all 8 patients.

including translocation t(4;14), deletion 17p and gain of 1q21 (all $p < 0.05$, chi-square test, Supplementary Data 3).

Since CXCL12 is a key chemokine in the pathogenesis of myeloma[17,18] we used ELISA and IHC to validate expression differences at the protein level. Of note, we did not find a consistent down-regulation of CXCL12 in sorted MM cells from focal lesions using ELISA ($n = 6$ patients, mean ratio RBM/FL: 1.26, (range: 0.91–1.75), $p = 0.46$ in linear mixed-effects model, Supplementary Fig. 5g). This was in line with findings from stained histology sections which did not show consistent differences in the proportion of CXCL12-positive MM cells (focal lesion median: 5 (0.04–20.6) vs. RBM median: 1.35 (0.044–20.1), $p = 0.64$ in two-sided Wilcoxon signed rank test, Fig. 3d). Due to limited patient material and low expression in MM cells, we could not assess protein levels of *CXCL7*.

One possible explanation for the difference between mRNA and protein levels of CXCL12 could be internalization of the protein via the receptors CXCR4 and CXCR7. For CXCR4, we found up to two-fold differences between paired samples from 11 patients using bulk RNA-seq but the changes were seen in both directions (7x down-and 4x upregulated in focal lesions, $p = 0.67$ in two-sided Wilcoxon signed rank test, Supplementary Fig. 5h). Similarly, there was no significant difference in the proportion of CXCR4-positive MM cells in IHC (focal lesion median: 41.04% (range: 2.4–91.6%) vs. RBM median: 65.1% (3.3–95.3%), $p = 0.20$ in two-sided Wilcoxon signed rank test, Supplementary Fig. 5i). While our findings for CXCR4 do not support increased internalization via this receptor, the CXCL12 scavenger-receptor CXCR7 showed a fivefold upregulation at the mRNA level in the paramedullary lesion of patient P13. This paramedullary lesion presented with a 215-fold down-regulation of the *CXCL12* gene compared to the RBM but showed similar values at the protein level according to IHC and ELISA (Fig. 3d and Supplementary Fig. 5g).

An alternative explanation for the difference in mRNA and protein levels could be increased supply of MM cells with CXCL12. Mesenchymal stromal cells (MSCs) have been described as an important source of CXCL12 for myeloma cells[19]. Using flow cytometry, we observed a significantly increased proportion of MSCs (CD271+, CD90+) in the TME of focal lesions in 8 patients with paired samples (mean RBM = 0.08% (range:0.004–0.23%) vs mean focal lesion = 2.50% (0.06–9.12%), $p = 0.04$ in two-sided Wilcoxon signed rank test, Supplementary Fig. 5j), indicating that a major source of CXCL12 is enriched in the TME of focal lesions.

Taken together, bulk RNA-seq and GEP data reveal consistent changes in expression profiles of focal lesions compared to paired random samples, including down-regulation of chemoattractant cytokines. Yet, the down-regulation of *CXCL12* could not be validated at the protein level, probably due to an enrichment of CXCL12-producing cells in the TME of focal lesions and internalization of this cytokine by MM cells.

## Tumor subclones show different levels of spatial transcriptional plasticity

The next question we addressed was if spatial differences as seen in bulk RNA-seq are due to the presence of unique subclones or the result of transcriptional or epigenetic plasticity. Therefore, we defined the transcriptional and chromatin accessibility profiles of genetically identical subclones at different BM sites. To avoid confounding due to small cell numbers we only considered CNA-defined subclones with >50 cells at both BM sites. All 6 genes, which we identified as differentially expressed between focal lesion/RBM pairs using bulk RNA-seq and GEP, were barely or not detectable using scRNA-seq (Supplementary Fig. 6a). In paired scATAC-seq of four patients, there was low coverage at the *CEBPD*, *CXCL7*, and *CXCL12* gene loci, while the other three genes showed no differential accessibility in regulatory regions between focal lesions and RBM (Supplementary Fig. 6b).

Performing a global gene expression and chromatin accessibility analysis of genetically identical subclones at different BM sites, we observed two patterns, including very similar profiles and pronounced spatial heterogeneity in expression and accessibility profiles. Very similar profiles of genetically identical subclones at different BM sites were seen in patients P01 & P05, with subclones being assigned to the same transcriptional cluster (Fig. 4a, Supplementary Fig. 7a). For one of the patients (P05) paired scATAC-seq data was available and in line with scRNA-seq genetically identical subclones from different BM sites were assigned to the same chromatin accessibility cluster (Supplementary Fig. 7b).

In contrast, pronounced spatial heterogeneity in epigenetic and transcriptional profiles of identical subclones at different BM sites was seen in patients P03 and P04 (Fig. 4b, Supplementary Fig. 7c). Especially subclones 1 and 6 in patient P03 displayed strong differences between the FL and RBM and they were assigned to different transcriptional and chromatin accessibility clusters, suggesting changes induced by the TME (Fig. 4b, Supplementary Fig. 7d). Differences in gene expression included an upregulation of MYC-target genes at the RBM-site as well as an upregulation of IFNγ and IFNα-pathway genes in the focal lesion, with the same spatial changes being seen in both subclones (Supplementary Data 4). The predicted motif activity of transcription factors, which regulate interferon, such as IRF4, 8 and 9 was increased at the focal lesion site, suggesting epigenetic changes to underlie upregulation of the interferon pathway (Fig. 4c).

Furthermore, we found the immunotherapy target *CD38* and genes coding for MHC I and II components (*CD74*, *HLA-B*, *HLA-C*) among the top differentially expressed genes between genetically identical subclones at different BM sites of patient P03 (Fig. 4b). To further delineate the mechanism underlying increased CD38 expression at the focal lesion site, we assessed chromatin accessibility at the regulatory elements of CD38. We observed a strong correlation

between the *CD38* promoter and a distal putative enhancer as well as a regulatory intronic element only in the focal lesion (Fig. 4d). In addition, the CD38 promoter peak overlapped with IRF4 peaks in published chromatin immunoprecipitation with sequencing (ChIP-seq) data for the MM cell line KMS12BM[20] (Fig. 4d), suggesting a link between increased IRF4 activity and overexpression of CD38.

Interestingly transcriptional plasticity of immunotherapy targets and MHC-components was not only seen between different bone marrow sites, but also between coexisting/competing subclones at the same BM site. For instance, in patient P01 MM cells with a del(16q11.2-q24.3) showed a higher expression of *CD74* as well as decreased expression of HLA-B, and HLA-E compared to MM cells without this deletion. Furthermore, the site-unique dominant subclone at the focal lesion overexpressed *MS4A1*, which encodes the immunotherapy target CD20 (Fig. 5a). In patient P03 subclone 5 showed the highest expression of both *CD38* and *TNFRSF17* (also known as *BCMA*) (Fig. 5b).

Since differential expression of the MHC II component *CD74* was seen in multiple comparisons, we next examined the correlation between its expression and clinical parameters. In 653 newly diagnosed patients, low (≤median) expression of *CD74* was associated with inferior OS (HR: 1.39 (1.25–1.52), $p = 0.01$, Cox regression and log-rank test), as well as increased PC infiltration ($p = 0.03$, $t$ test) and higher rISS stages ($p = 0.047$, chi-square test, Supplementary Fig. 5f, Supplementary Data 3).

Overall, we observed different levels of spatial transcriptional and epigenetic plasticity, suggesting that both the presence of unique subclones as well as plasticity can contribute to expression differences between BM sites. Our results also indicate that genes coding for MHC components and immunotherapy targets can be differentially expressed both at the subclonal level and at different BM sites.

## Macrophages are depleted in focal lesions

Intratumor heterogeneity was recently associated with diversity of the TME in melanoma[21]. To gain a first insight into the composition of the cellular TME in focal lesions, we performed scRNA-seq of paired CD138-depleted mononuclear cell fractions with the focal lesion sample originating from intramedullary components. In total, 31,216 cells from 6 patients passed quality control. We integrated them into a single dataset and used an annotation, which was developed by Stuart and co-workers[22]. Splitting the data by sample type, we found a significant depletion of CD14- and CD16-positive monocytes/macrophages and their progenitors in the TME of focal lesions compared to the RBM (>2.5-fold, $p < 0.05$ in two-sided Wilcoxon signed rank test, Fig. 6a, b, Supplementary Data 5). For the T-cell compartment we observed a significant enrichment of CD8 Effector 2 T-cells in focal lesions (mean focal lesion=11.13% (1.7–29.6%) vs. mean RBM = 4.9% (1.6–10.4%), $p = 0.03$ in two-sided Wilcoxon signed rank test), however a more than twofold enrichment was only found in 3 patients (Supplementary Data 5).

The relative depletion of macrophages in the TME of focal lesions is a surprising finding, given their role as one of the key interaction partners of MM cells in the BM niche and the association between high numbers of macrophages in the TME and increased MM cell proliferation[18,23–25]. To validate our finding, we stained core biopsies from 21 patients (6 patients with scRNA-seq), and determined the proportion of macrophages, CD4- and CD8-positive T-cells (Fig. 6c, d). Unexpectedly, we observed a high intra-sample heterogeneity in proportions of macrophages, with both focal lesion and RBM specimens showing regional differences. To account for that, we quantified macrophages and T-cells at multiple sites within the biopsies and defined the MM infiltration at these sites in four intervals (1–25%, 26–50%, 51–75%, and 76–100%). We fit linear mixed-effects models for cell proportions and included the type of BM sample (RBM vs. focal lesion) and tumor load at investigated sites as fixed effects and patient ID as random effect. Details

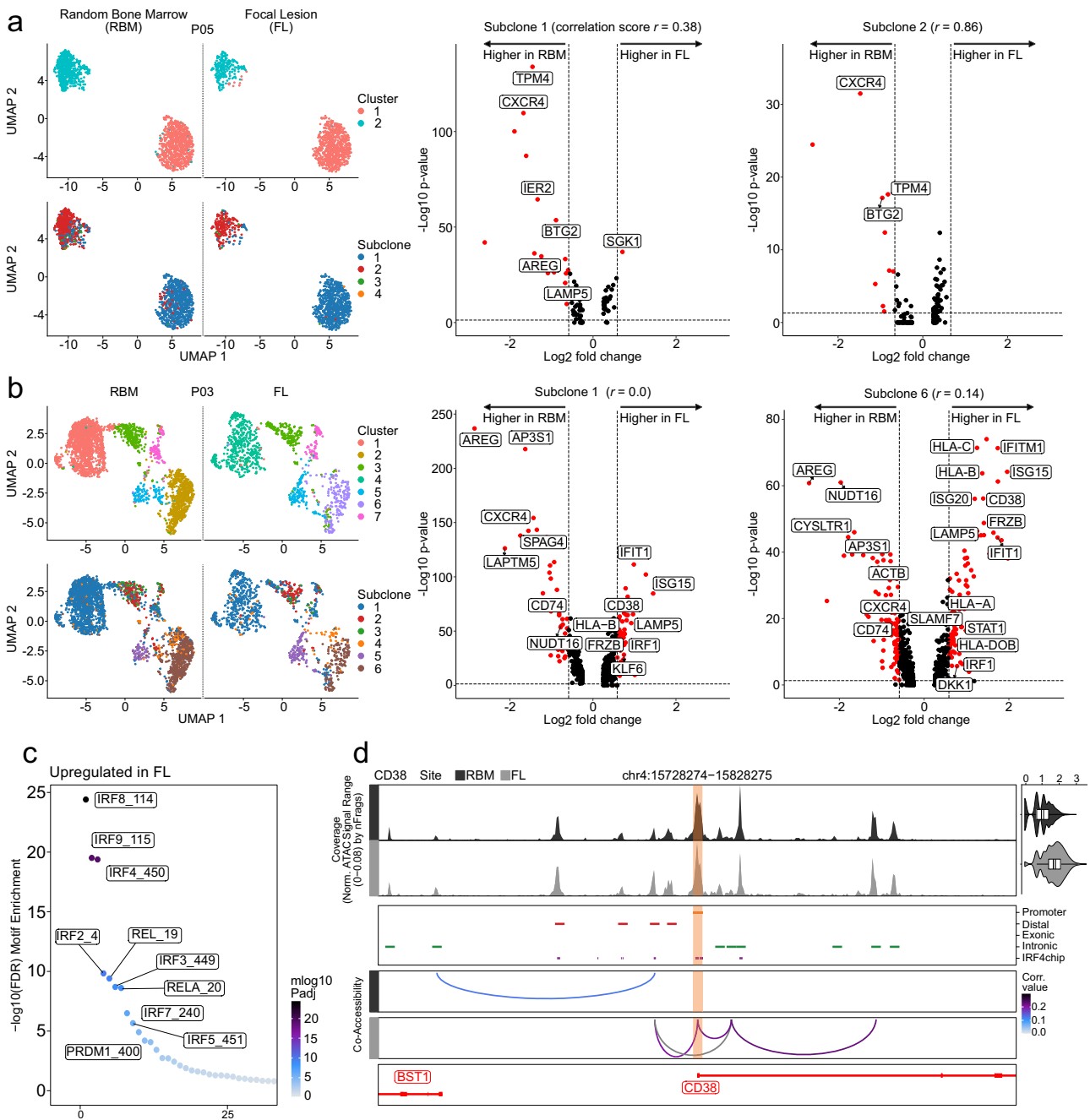

**Fig. 4 | Spatial transcriptional and epigenetic plasticity in tumor subclones.**
**a** Single-cell RNA-seq data for patient P05 as an example for very similar expression profiles of genetically identical subclones at different bone marrow sites. Left panel: transcriptional clusters and copy number aberration (CNA)-defined subclones, showing that subclones 1 and 2 were assigned to the same transcriptional clusters at the focal lesion (FL) and the random bone marrow (RBM) site. Right panel: volcano plot for the comparison of gene expression of these subclones in FL vs. RBM. Significant genes are highlighted and depicted in red (two-sided Wilcoxon rank sum test, Benjamini Hochberg adjusted *p* value < 0.05, ≥1.5-fold enrichment). In (**b**) the same plots as in (**a**) are shown for patient P03 as an example for pronounced differences between genetically identical subclones at different bone marrow sites. The two subclones 1 and 6 were assigned to different transcriptional clusters at the FL and the RBM, suggesting differential expression of the same subclone at different bone marrow sites. **c** Comparison of transcription factor (TF) motif deviation

scores between paired samples of patient P03. The top TFs of the FL are marked. Marker peaks were identified based on two-sided Wilcoxon rank sum test.
**d** Chromatin accessibility at the CD38 promoter plus 50000 bps upstream and downstream in paired samples of patient P03. The CD38 promoter peak is highlighted in light orange. Top panel: aggregated pseudo-bulk scATAC-seq tracks for the RBM (dark gray) and the FL (light gray). Right panel: violin plots showing normalized CD38 expression from scRNA-seq data per spatial site (RBM: 824 cells, FL: 193 cells). The boxplots show the median and the interquartile range, while the upper and lower whiskers show the highest and lowest value. Middle panel: peaks are colored based on the location of the peak in either promoter (orange), distal (red), exonic (blue) or intronic (green) regions. IRF4 ChIP-seq peaks from the multiple myeloma cell line KMS12BM are shown in purple; Bottom panel: peak co-accessibility in the CD38 region at both bone marrow sites colored by Pearson correlation coefficient.

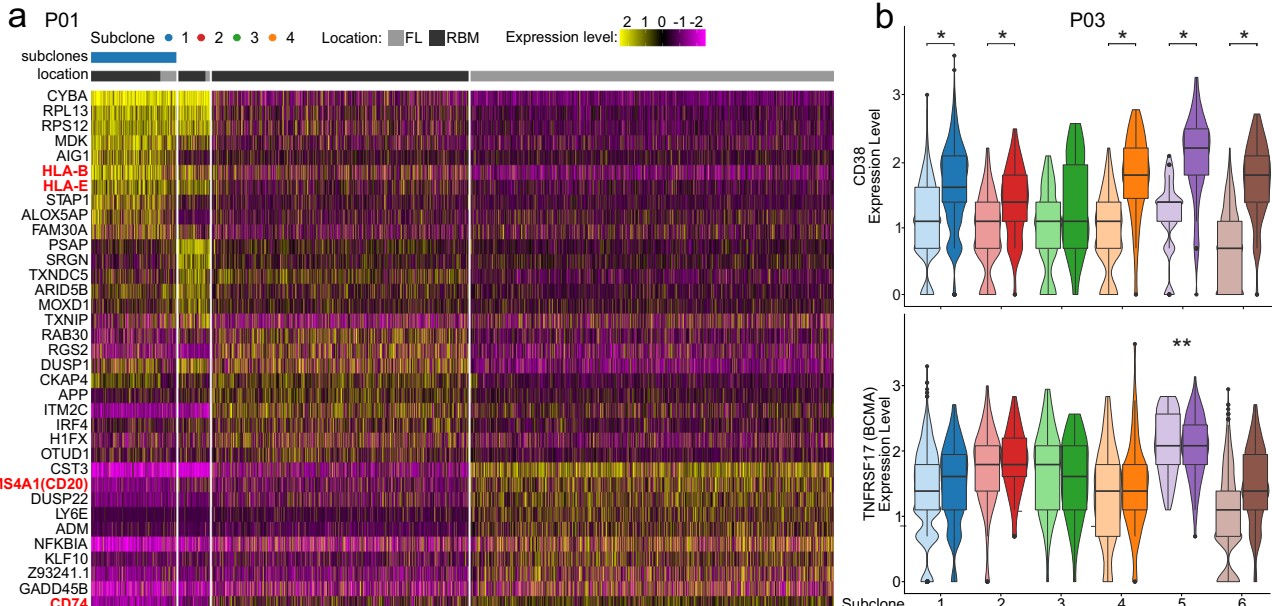

**Fig. 5 | Transcriptional plasticity of immunotherapy targets and MHC-components between coexisting/competing subclones. a** Heatmap showing the top significantly differentially expressed genes per subclone of patient P01. Over-expression of HLA-genes in subclone 1, upregulation of *MS4A1 (CD20)* in the focal lesion-unique subclone 4, and increased expression of *CD74* in subclones 3 and 4 are highlighted in red. **b** Violin plots for the immunotherapy targets *CD38* and *TNFRSF17 (BCMA)* expression in subclones of patient P03 at the random bone marrow (RBM, *light* color) and focal lesion (*FL, dark* color) site (subclone 1: 1520

RBM|629 FL cells, 2: 157|114 cells, 3: 97|34 cells, 4: 153|54 cells, 5: 69|125 cells, 6: 695| 183 cells). *P* values were calculated using two-sided Wilcoxon rank sum tests with Benjamini Hochberg adjustment. *Significant overexpression (*p* < 0.05) at the focal lesion site. **The highest levels of *TNFRSF17 (BCMA)* expression were seen in sub-clone 5 at both the focal lesion and random bone marrow site. The boxplots show the median and the interquartile range, while the upper and lower whiskers show the highest and lowest value.

of the models are shown in Supplementary Table 2. There were significantly more macrophages in RBM specimens compared to focal lesions (+8%, standard error (SE) 1.20, *p* < 0.0001). We also found a significant, albeit smaller difference for CD8-positive T-cells (+3.86% in RBM, SE 1.14, *p* = 0.0006). For CD4-positive T-cells there was no significant difference between RBM and focal lesions (*p* > 0.05). Of note, there was a strong decline in macrophages from regions with a low MM cell infiltration (<25%, interval 1) compared to sites with a high infiltration (>75%, interval 4) (Fig. 6d). The respective estimated proportions were 41.04% (SE: 2.25) in interval 1 and a drop of −40.25% (standard error (SE) 1.93) in interval 4. The decline was less pronounced for CD4-positive (22.99% (SE: 2.02) in interval 1 & −21.49% (SE: 1.95) in interval 4) and CD8-positive T-cells (17.88% (SE: 2.23) in interval 1 & −14.98% (SE: 2.02) in interval 4). Importantly, this finding could also explain the relative depletion of macrophages in focal lesions, since they had a significantly higher plasma cell infiltration compared to the paired RBM (mean focal lesions=62.0%, (range:10.1–96.8%) vs. mean RBM = 32.2% (0.8-74.7%), *p* = 0.001 in two-sided Wilcoxon signed rank test, Fig. 6e), and we cannot exclude that this also holds true for sites with the highest infiltration (interval 4).

To further validate changes in the TME, we performed flow cytometry, including paired samples from 8 patients. There was a trend towards a lower proportion of monocytes/macrophages in the TME of focal lesions (mean focal lesion=8.01% (range: 2.35–20%) vs. mean RBM = 11.36% (6.58–25.33%), *p* = 0.078 in two-sided Wilcoxon signed rank test, Supplementary Fig. 8a). Supporting the link between tumor load and macrophages, the two patients who had almost the same plasma cell infiltration in the paired samples (P11 and P18 in Supplementary Fig. 8b) showed no depletion of monocytes/macrophages.

Similar to macrophages, we observed a depletion for CD4-positive T-cells (mean focal lesions = 7.28% (range:1.89–12.90%) vs. mean

RBM = 9.95% (3.71-26.94%), *p* = 0.16 in two-sided Wilcoxon signed rank test). In contrast, for CD8-positive T-cells there was a trend towards higher proportions in the focal lesions (mean focal lesion=20.23% (5.01-73.91%) vs. mean RBM = 9.42% (4.97–18.29%), *p* = 0.06 in two-sided Wilcoxon signed rank test, Supplementary Fig. 8c, d).

Taken together, using scRNA-seq we observed a significant rela-tive depletion of macrophages in the TME of focal lesions. Validation experiments using IHC and flow cytometry suggest that the depletion is strongly associated with the level of plasma cell infiltration rather than being a unique feature of focal lesions.

**The T-cell repertoire displays spatial differences with regard to clonal proportions**
T-cells are the main contributor to the adaptive anti-tumor response by recognizing tumor neoantigens through their T-cell receptor (TCR)[26]. To address the questions if the T-cell repertoire is heterogeneous between BM sites and if there are site-unique T-cell clones in MM patients, we combined scRNA-seq of the TME with TCR-sequencing (TCR-seq) (Fig. 7). Since the BM regularly contains T-cells, and clonal expansion of T-cells has been linked to tumor reactivity[26,27], we focused on expanded (proportion ≥1%) and hyperexpanded (proportion ≥5%) T-cell clones. TCRs could be assigned to 11467 of 13702 (84%) T-cells from 6 patients. We found expanded T-cell clones in all patients (median: 10 clones, range: 2–16). At least one hyperexpanded clone was found in 3 patients (range: 1–3) (Fig. 7a, Supplementary Data 6). Expanded T-cell clones almost exclusively resided in the CD8-positive compartment, with 75% (range 0–89%) and 16% (range 0–51%) of them being CD8 Memory 2 cells (CD45RO[+]/CD57[+]) and CD8 Effector 2 cells (CD69[+]), respectively (Fig. 7b, Supplementary Data 7). In line with this observation, we found a strong positive correlation between the number of expanded T-cell clones and the proportion of CD8-positive T-cells for both focal lesion (Pearson correlation 0.90, *p* = 0.01) and RBM (Pearson correlation 0.97, *p* = 0.001) (Supplementary Fig. 8e).

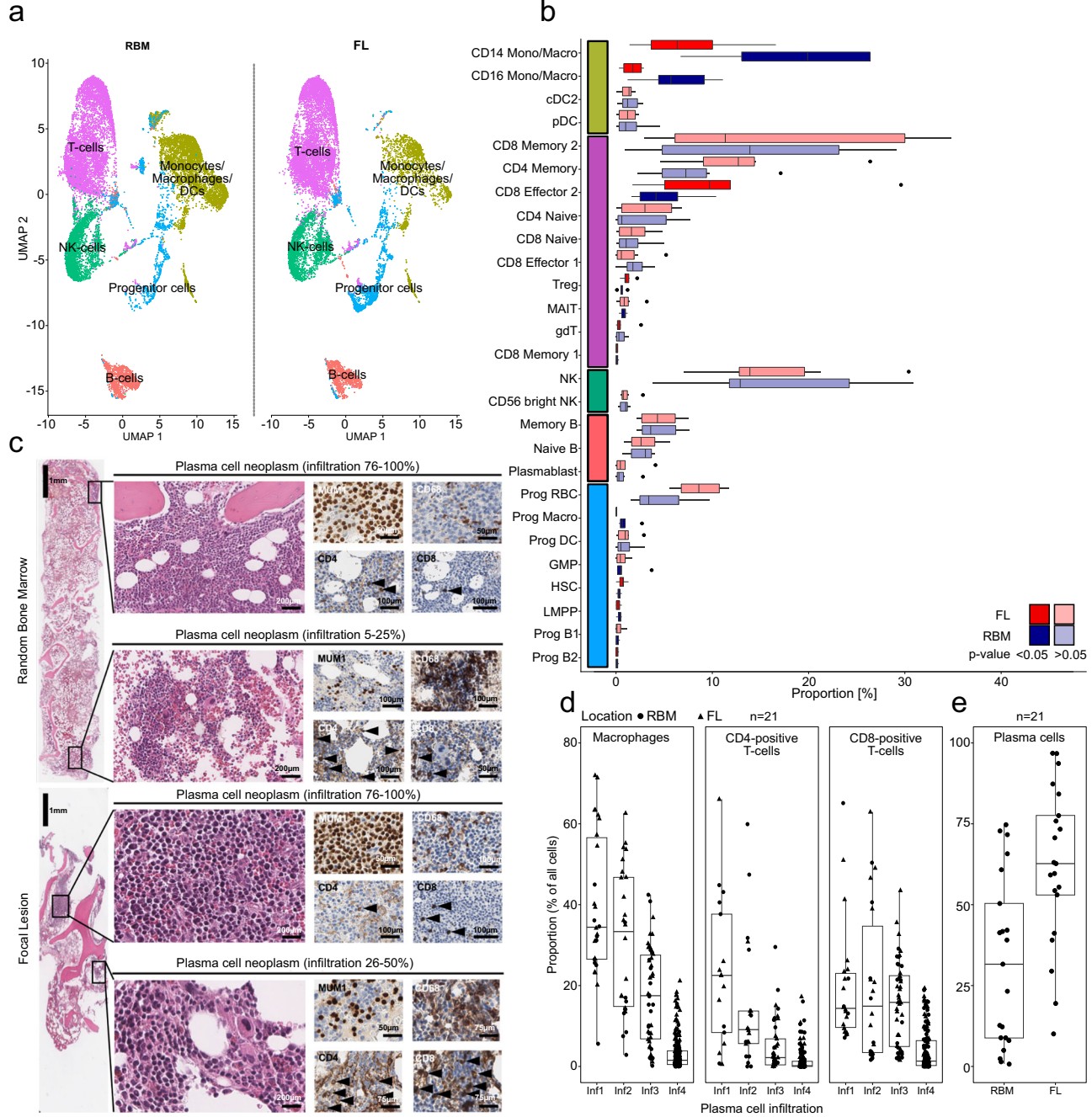

**Fig. 6 | Spatial heterogeneity of the tumor microenvironment. a** Uniform Manifold Approximation and Projection (UMAP) of scRNA-seq for paired focal lesion (FL) and random bone marrow (RBM) CD138-negative mononuclear cell fractions from 6 patients (P01-P06). The exact cell numbers per boxplot are shown in the Source Data. **b** Boxplots for the proportion of individual cell types in FL (red) and RBM (blue) specimens from the 6 patients. Dark colors indicate significant (*p* < 0.05) differences between FL and RBM, which were assessed using two-sided Wilcoxon signed-rank tests. The boxplots show the median and the interquartile range, while the upper and lower whiskers show the highest and lowest value. The colors on the y-axis correspond to the main cell types in (**a**). Mono/Macro monocytes/macrophages; cDC2 conventional dendritic cells (DC) 2, pDC plasmacytoid DC, Treg Regulatory T-cells, MAIT mucosal-associated-Invariant T-cells, gdT gamma/delta T-cells, Prog progenitor, RBC red blood cells, Macro macrophage, GMP granulocyte monocyte progenitors, LMPP lymphoid-primed-multi-potential

progenitor cells. **c** Evaluation of cellular proportions using immunohistochemistry (IHC). Stainings of FL and RBM specimens from patient P01 are shown. Slides were stained for MUM1 (plasma cells), CD68 (macrophages), CD4 (CD4-positive T-cells), or CD8 (CD8-positive T-cells). Representative regions with low and high plasma cell infiltration are shown. **d** Boxplots for the proportion of macrophages, CD4- and CD8-positive T-cells according to IHC for 21 patients with paired RBM/FL samples. Regions in stained slides were classified according to the level of plasma cell infiltration: 1–25% (Inf1), 26–50% (Inf2), 51–75% (Inf3) and 76–100% (Inf4). Multiple regions with a different level of plasma cell infiltration were considered for each slide. **e** Boxplot for the plasma cell infiltration in paired RBM and FL from the same 21 patients according to IHC. The boxplots show the median and the interquartile range, while the upper and lower whiskers show the highest and lowest value (excluding outliers), respectively. Source data are provided as a Source Data file.

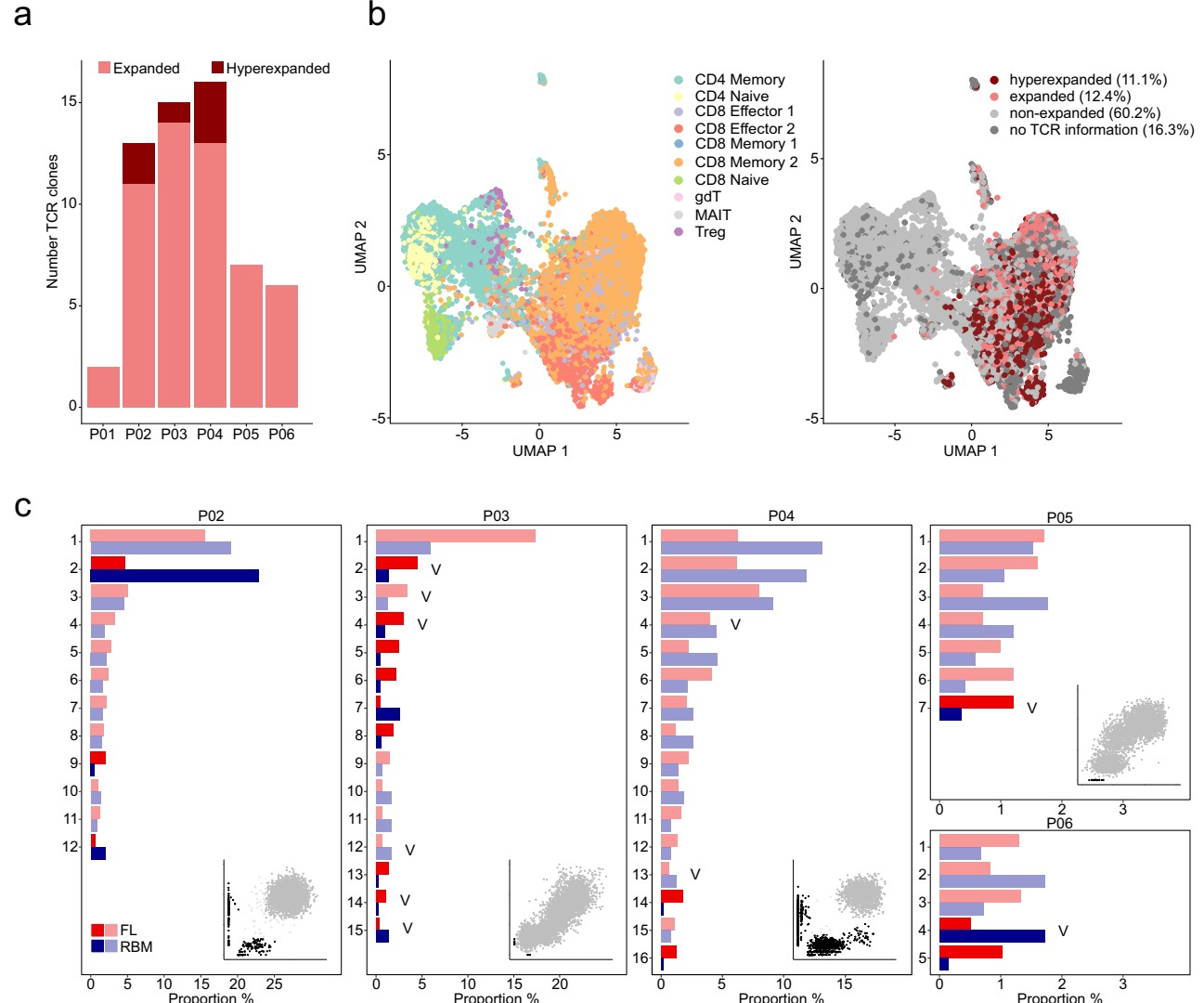

**Fig. 7 | The spatial architecture of the T-cell repertoire. a** Total number of detectable expanded (proportion: 1–5%, light red) and hyperexpanded (>5%, dark red) T-cell clones per patient in CD138-negative mononuclear cell fractions. **b** Uniform Manifold Approximation and Projection (UMAP) of the T-cell compartment with 10 different subtypes (left panel), and T-cell receptor (TCR) information mapped into the UMAP space (right panel). Colors indicate the type of T-cell expansion. **c** Proportion of expanded and hyperexpanded T-cell clones in paired focal lesion (FL)/random bone marrow (RBM) samples, which are shown as red and blue bars, respectively. Since only T-cell clones with at least 10 cells in one of the

paired samples were considered for the comparison, data for patient P01 is not depicted. Except for patient P06 (no whole genome sequencing data available), the cancer clonal fraction plot for total single nucleotide variants (SNVs) in paired RBM (x-axis) and FL (y-axis) was added to show the extent of spatial tumor heterogeneity. Unshared and site-enriched SNVs are marked in *black* and shared SNVs in gray. For the TCR clones marked with a "V", the CDR3 was associated with recognition of epitopes derived from the viruses CMV, EBV or Influenza A based on the databases VDJdb[37] and McPas-TCR[38]. Source data are provided as a Source Data file.

Using Wilcoxon signed rank tests, we did not observe significant differences in exhaustion and cytotoxicity scores[28] between expanded and non-expanded T-cell clones in our set (Supplementary Fig. 8f), and the same holds true for expanded T-cell clones in focal lesions vs. RBMs (Supplementary Fig. 8g).

For the comparison of T-cell repertoire between paired samples, we only included T-cell clones with at least 10 cells at one BM site to avoid an overestimation of heterogeneity. Using this stringent criterion, none of the expanded T-cell clones were site-unique and expansion was typically seen in both paired samples. Using a threefold difference as an arbitrary cut-off, at least one expanded clone (range: 1–9) showed variation in focal lesion/RBM pairs (Fig. 7c). Yet, changes in T-cell clone proportions were also seen in paired samples from patient P03 with hardly any differences in genomic profiles between focal lesion and RBM (Fig. 7c). We conclude that proportional changes in the T-cell repertoire are not necessarily linked to genetic

heterogeneity. In line, searching curated databases of TCR sequences[29,30], CDR3 sequences from a total of 10 expanded T-cell clones in our set were associated with recognition of epitopes derived from viruses (CMV, EBV or Influenza A), including 6 T-cell clones with >3-fold spatial differences (Fig. 7c).

In summary, we observed proportional changes in the T-cell repertoire in paired samples. Yet, in our set none of the expanded T-cell clones was site-unique and there were no consistent spatial differences in exhaustion or cytotoxicity scores.

## Discussion

MM is a heterogeneous disease, in which focal lesions are associated with poor outcome[11,12]. To define the subclonal architecture and expression signatures as well as the composition of the TME in focal lesions in detail, we applied a comprehensive approach including bulk WGS, RNA-seq as well as the two sc-sequencing techniques, scRNA-seq

and scATAC-seq. Using this advanced approach, we describe expression differences between unique subclones and show that genetically identical subclones can display different levels of spatial transcriptional plasticity, including nearly identical profiles and pronounced heterogeneity at different bone marrow sites. We demonstrate a depletion of macrophages at sites with a high plasma cell infiltration and proportional changes in the T-cell repertoire in intramedullary focal lesions.

In our prospective study, we observed spatial heterogeneity at a similar extent as seen in a recent whole-exome sequencing study, including site-unique tumor subclones[5]. At first glance, presence of site-unique or strongly enriched tumor subclones is in contrast to another multi-region scRNA-seq analysis in MM which provided only limited evidence for spatial genomic heterogeneity in newly diagnosed patients[31]. Merz and co-workers primarily used samples from small osteolytic lesions to explore the mechanism underlying bone disease. Yet, Rasche et al. recently demonstrated that spatial heterogeneity was associated with the size of focal lesions[5,11], and consequently we focused on focal lesions with a diameter >1 cm, which could rationalize the observed differences.

Rasche and co-workers proposed that the evolutionary selective pressure in the BM niche could lead to the selection of MM subclones with decreased BM dependence[5]. Our findings on the depletion of macrophages in the TME provide evidence for this hypothesis. Macrophages support cancer hallmarks in MM, such as angiogenesis[32] and prevent MM cell death through IL-6/STAT3 signaling and BAFF/BCMA-interactions[25,33]. Hence, the depletion of macrophages in focal lesions suggest that MM cells at these sites are less dependent on one of the key plasma cell interaction partners in the BM. Of note, the number of macrophages has been shown to be associated with more aggressive disease, worse prognosis and treatment resistance in MM[34,35]. Thus, our findings have potential implications for models of MM disease progression. Furthermore, they might also impact the development of macrophage targeting therapies in MM and the response of focal lesions to immunotherapy which requires antibody dependent phagocytosis[36].

We appreciate that further research is required to conclude if the depletion of macrophages is the cause or the result of the expansion of advanced subclones in focal lesions and if they have any impact on therapy response and tumor cell expression profiles. The same holds true for changes in tumor cell expression profiles, including downregulation of genes coding for chemokines such as *CXCL12*. We could not validate this finding at the protein level but our preliminary findings on changes in myeloma cells and the TME provide potential explanations for this discrepancy. De Jong et al. recently demonstrated the important role of CXCL12-producing inflammatory mesenchymal stromal cells in the pathogenesis of MM[19]. Here, we show a significant increase in the proportion of mesenchymal stromal cells in the TME of focal lesions and upregulation of the CXCL12-scavenger receptor CXCR7 in the patient with the strongest down-regulation of CXCL12 at the mRNA level, indicating increased supply with CXCL12 and/or uptake of the chemokine by tumor cells in focal lesions. We appreciate that this finding needs to be confirmed in larger studies.

While we identified changes in expression profiles between focal lesions and paired RBM, with our limited sample set we could not address the question if there are transcriptional signatures which discriminate between site-unique and disseminated tumor subclones to gain deeper insights into "myeloma metastasis" and/or focal lesion formation. Thus, we propose a spatial-longitudinal study, starting at premalignant stages of the disease, with samples from different intra- and extramedullary sites to better understand the complex evolutionary processes that result in BM independence and more aggressive

disease. However, we need to emphasize that lesions are often not accessible or only with a risk to the patient that is not acceptable for a purely diagnostic study.

A spatial-longitudinal study would also improve our understanding why focal lesion subclones, which are characterized by unique mutations, can expand in the BM, without triggering an efficient immune response. Yet, our preliminary findings on the T cell repertoire at different BM sites already provide some clues. Expansion of T-cell clones has been described as a marker for tumor specificity in melanoma[26,27]. While we observed quantitative differences in the T-cell repertoire, we did not detect site-unique expanded T-cell clones in our set of intramedullary focal lesions, which is in contrast to metastasis infiltrating T-cells in melanoma[21]. Thus, increased immune dysfunction could be one potential mechanism underlying the ability to expand. While even non-expanded T-cell clones can be tumor specific[27], we provide further evidence for escape of immune surveillance, including differential expression of MHC-I and MHC-II components between subclones and/or BM sites. Yet, we appreciate that the failure to show a link between tumor genomics and expanded T-cell clones could be due to the limited sample size of our study.

Strikingly, we also observed location and subclone-specific epigenetic and transcriptional plasticity for genes coding for key immunotherapy targets such as BCMA and CD38. This suggests that at least in some patients subclones and focal lesions could potentially respond differently to immunotherapy[37,38]. We recently provided clinical evidence for this assumption: using functional imaging for response assessment in a patient treated with the monoclonal anti-CD38 antibody daratumumab, we observed both responding and progressing focal lesions[39]. However, we cannot exclude that expression differences were caused by subclonal differences in stress responses induced by sample processing. We also appreciate that this finding needs to be validated on the protein level, which was not possible in our study due to limited biomaterial.

In conclusion, our study reveals more insights into spatial heterogeneity in myeloma, including the subclonal structure in focal lesions and the processes underlying increasing independence of MM cells from the BM microenvironment. Further research is justified to resolve the next stage of focal lesion evolution in MM, such as para- and extramedullary disease, and the role of focal lesions in a relapsed setting, which is likely to provide even deeper insights into the pathogenesis of MM.

## Methods
### Patients
This study was approved by the Heidelberg University Medical Faculty ethics review board (S278-13) and complied with all relevant ethical regulations. Signed written informed consent for sample procurement and processing in accordance with the Declaration of Helsinki was obtained for all cases included in this study. Study subjects did not receive compensation for participation in the study. For the comparison of paired samples, we included 31 Caucasians with newly diagnosed myeloma and accessible focal lesions. Sex was not considered in the initial study design due to the small number of eligible patients. All patients fulfilled the International Myeloma Working Group (IMWG) criteria for treatment[40]. CD138-enriched MM cells for molecular analyses were available from 17 of these patients. Patient clinical characteristics, the origin of samples and the analyses, which were performed with the respective samples, are shown in Fig. 1b and Supplementary Tables 3, 4. For correlation analyses between gene expression and patient characteristics as well as outcome, we included 653 newly diagnosed patients from two phase III clinical studies (GMMG HD4 and MM5 trials) with available gene expression profiling data[41].

## Medical imaging

Whole-body [18]F-Fluorodeoxyglucose (FDG)–positron emission tomography (PET) computer tomography (CT) was performed with a Biograph mCT, S128 (Siemens Co., Erlangen, Germany) 1 h after injection of [18]F-FDG. For attenuation correction of PET data and image fusion, a low-dose attenuation CT (120 kV, 30 mA) was used. An image matrix of $400 \times 400$ pixels was used for iterative image reconstruction, which was based on the ordered subset expectation maximization (OSEM) algorithm with two iterations and 21 subsets as well as time of flight (TOF). The reconstructed images were converted to SUV images based on the formula: SUV = tissue concentration (Bq/g)/(injected dose (Bq)/body weight (g)).

Magnetic resonance imaging (MRI) was performed with a 1.5 Tesla MRI Scanner (Siemens Co., Erlangen, Germany). Imaging sequences comprised coronal T1-weighted turbo spin echo (T1w; repetition time 528 ms, echo time 8.4 ms, in plane resolution 1.3 mm × 1.3 mm, slice thickness 5.0 mm, 10% distance factor), coronal T2 weighted short-TI inversion recovery (T2w; repetition time 3650, echo time 56 ms, in plane resolution 0.7 mm×0.7 mm (interpolated), slice thickness 5.0 mm, 10% distance factor, fat suppression: slice-selective inversion recovery (TI = 160 ms)), and axial diffusion weighted imaging (DWI; Diffusion-EPI iShim, repetition time 5130, echo time 64 ms, in plane resolution 1.8 mm×1.8 mm (interpolated), slice thickness 6.0 mm, 0% distance factor, b values: 50 s mm$^{-2}$ and 800 s mm$^{-2}$, fat suppression: slice-selective inversion recovery (TI = 180 ms)).

For PET, a focal lesion was defined as a circumscribed focus with increased FDG uptake compared with its surroundings. For MRI, focal lesions were defined as focal hypointensity in T1 and as hyperintensity in T2 and the high b-value image from DWI. The median diameter of the analysed focal lesions was 2.6 cm (range: 1.4–7.8 cm). Guided sampling was performed using a Siemens Emotion 16 CT (Siemens Co., Erlangen, Germany) or by surgical resections. The number and size of osteolytic lesions is given in Supplementary Data 8.

## Sample preparation

Focal lesions for CT-guided biopsies were chosen according to accessibility and minimal risk for the patient. In case of multiple eligible lesions, the best accessible lesion was selected. CT-guided biopsies were performed in a separate session within one week after collection of the sample from the iliac crest. Biopsies were performed prior to treatment. For each site 10–15 ml of BM were collected and processed on the same day. Mononuclear cells (MC) from random BM and intramedullary focal lesion aspirates as well as from peripheral blood (PB) were isolated using the Ficoll–Paque method. Subsequently, BM MM cells were enriched using an immuno-magnetic CD138-positive selection (Robosep, Stemcell Technologies). The CD138-positive and the CD138-negative BMMC fraction were either stored in Qiagen RLT buffer (bulk sequencing) at −80 °C or viably frozen in dimethylsulfoxide at a final concentration of 10% (sc sequencing) at −150 °C. PBMCs were stored as dry pellets at −20 °C. For patient P09, we isolated DNA for whole-genome sequencing from a fresh frozen specimen. Samples from surgical resections of paramedullary lesions were minced and enzymatically digested for 15 min at 37 °C in MEM alpha Medium (Gibco, USA) containing 1 mg/ml Collagenase II, 0.8 mg/ml Dispase (both Gibco, USA) and 0.1 mg/ml DNAse I (SigmaAldrich, USA). Cells were released by pipetting repetitively (30x). The reaction was stopped using a quenching buffer (Calcium-free PBS + 5 mM EDTA + 2% FCS). For sequencing, MM cells were sorted (CD38 high, CD45RA-) using a FACSAria (BD Biosciences). Dead cells were identified and excluded using eFluor-506 (ThermoFisherScientific, USA). Sorted cells were stored in RLT buffer (Qiagen, Hilden, Germany) before bulk sequencing. The type of processing is shown for each sample in Supplementary Table 5.

## Flow cytometry

Cryopreserved samples were thawn at 37 °C and washed twice in ice-cold 1x PBS. Used antibodies are shown in Supplementary Table 6. Measurements were performed on a FACSymphony (BD Biosciences). Dead cells were excluded using eFluor-506 (ThermoFisherScientific, USA). The gating strategy for the identification of plasma cells (CD38+, CD138+), monocytes/macrophages (CD11b+, CD33+), MSCs (CD90+, CD271+), CD8 + T-cells (CD3+, TCRab+, CD8+) and CD4 + T cells (CD3+, TCRab+, CD8−) is shown in Supplementary Fig. 9. Only samples with more than 10,000 cells were included.

## Whole genome sequencing and variant calling

DNA of CD138-positive fractions from RBM and focal lesion samples as well as the corresponding PB (germline control) was isolated using the Allprep Kit (Qiagen). WGS libraries were prepared with the Illumina TruSeq Nano DNA kit and sequenced on a HiSeq X (paired-end 150 bp), average coverage 85x for tumor and 43x for germline control samples. Raw sequencing data was processed and aligned to human reference genome build 37 version hs37d5 using the DKFZ OTP WGS pipeline[42]. Copy number aberrations (CNAs) were identified using ACEseq (v1.2.8-4)[43], and single nucleotide variants (SNVs) using samtools mpileup (v1.2.166-3)[44]. For the SNVs additional filtering steps were applied, including blacklist filtering[45], fpfilter (https://github.com/genome/fpfilter-tool) and removal of SNVs located in regions coding for immunoglobulins. For SNVs, which were only called in one of the paired samples, Rsamtools (v2.6.0) was used to determine the number of reference and variant reads in both samples. Manual somatic variant refinement using IGV (v2.7.2)[46] was performed according to a published standard operating procedure[47]. The cancer clonal fraction (CCF) was calculated using the following equation[48]:

$$n_{mut} = f_s \times \frac{1}{p} \left[ p n_{locus}^t + 2(1-p) \right] \tag{1}$$

where $n_{mut}$ is the mutation copy number, $f_s$ is the fraction of mutated reads (variant allele frequency), $p$ is the tumor purity, and $n_{locus}^t$ is the locus-specific copy number. Tumor purity $p$ was estimated based on histograms for the variant allele frequency (VAF) of SNVs (purity values are shown in Supplementary Table 5). For $n_{locus}^t$ we used the values predicted by ACEseq. We then compared the expected $f_s$ value to values assuming the mutation was on 1,2,3, …, C chromosomes and assigned $n_{chr}$ the value of C with the maximum likelihood using a binomial distribution. Finally, the CCF was determined by dividing $n_{mut}$ by $n_{chr}$. To avoid an overestimation of heterogeneity at the mutational level, only SNVs with a CCF > 0.15 in at least one of the paired samples were considered for downstream analyses. Aberrations with the same or similar CCF in paired samples were called *shared*[5]. Aberrations, which were detectable in only one of the paired samples, were called *unshared*. Furthermore, we discriminated between minor and major events: We calculated 95% confidence intervals (95% CI) for CCFs and classified mutations as major, if the upper band of the 95% CI was ≥ 1, and minor otherwise[49]. Furthermore, major mutations with a 3-fold enrichment between the paired samples were classified as enriched. Only nonsynonymous SNVs with a CADD score[50] > 20 were considered for driver gene[6] analyses. For CNAs, we used a CCF cut-off of 0.6 to discriminate between minor and major events.

## Gene-expression profiling (GEP)

GEP of CD138-enriched BM plasma cells was performed using Affymetrix U133Plus2.0 microarrays (Santa Clara, CA) according to the manufacturer's instructions. As chip definition file (CDF) we used the Affymetrix U133 Version 2.0 plus array custom (CDF) mapping to Entrez genes (http://brainarray.mhri.med.umich.edu/Brainarray/Database/CustomCDF/). Expression data were normalized using GC-RMA and converted to log2 scale.

## Bulk RNA-sequencing

RNA-Seq libraries were prepared using the Illumina TruSeq stranded mRNA kit and sequenced on the Illumina NovaSeq 6000 PE 100 S1 platform. Paired-end reads were mapped to the STAR index generated reference genome (build 37, hs37d5) using STAR v2.5.2b[51]. Gene expressions were quantified using featureCounts (Subread v1.5.1). Differential gene expression analysis was performed with the R-package DESeq2 (v1.28.1)[16] using tumor purity as covariate. Regularized logarithm (rlog) function was used for transformation to log2 scale and normalization including library size. Differentially expressed genes were visualized using the R-package EnhancedVolcano (https://github.com/kevinblighe/EnhancedVolcano, v1.6.0).

## Single-cell RNA-sequencing including VDJ-sequencing

Cryopreserved samples were thawed at 37 °C and washed twice in ice-cold 1x PBS. For CD138+ and CD138- BMMCs single-cell RNA plus B and T-cell receptor sequencing were performed using the Chromium Next GEM Single Cell 5' Reagent Kit v1.1 and the V(D)J Reagent Kit v1.1 according to the manufacturer's protocol (14,000 cells per channel). Generated gene expression libraries were paired-end sequenced on the NovaSeq 6000 S2. Generated V(D)J libraries were paired-end sequenced on the NextSeq 550.

## Single-cell ATAC-sequencing

Viably frozen CD138+ BMMCs were thawed and washed once with 1x PBS. Cell pellets were carefully resuspended in an ice-cold NP-40 lysis buffer (10 mM Tris-HCl, pH 7.4, 10 mM NaCl, 3 mM MgCl2, 0.1% IGEPAL CA-630) and spun down immediately. Nuclei were resuspended in the 10X Genomics nuclei buffer, counted and subjected to Tn5 tagmentation. The subsequent steps were done according to the manufacturer's instructions for the 10X Genomics Single Cell ATAC v1.0 or v1.1 Kit. Generated scATAC-seq libraries were paired-end sequenced on a NovaSeq 6000 S2.

## Preprocessing and analysis of single-cell RNA-sequencing data

For each demultiplexed library CellRanger count (v5.0.0) was run with reference refdata-gex-GRCh38-2020-A to quantify single cell feature counts. Cellranger vdj (v5.0.0) was used for each single TCR and BCR library (with reference refdata-cellranger-vdj-GRCh38-alts-ensembl-5.0.0). The count matrices were loaded into R (v4.0.2) by using the standard Seurat (v4.0.1) parameters and annotated for patient, location, sorting fraction and then merged. Cells with more than 5% mitochondrial RNA, less than 200 or more than 5000 expressed genes were removed (Supplementary Fig. 3a, Supplementary Table 7). Cell doublets were removed using Scrublet[52] (prediction score >0.3). Immunoglobulin genes were removed. Normalization was done using SCtransform[53] and technical or biological confounding effects as mitochondrial counts or cell cycle stages were regressed out using the "vars.to.regress" argument. Harmony (v1.0)[54] was used for integration of samples. Samples were clustered together based on Seurat k-nearest neighbors clustering with a resolution of 0.5. Cells were embedded into a two-dimensional space using Uniform Manifold Approximation and Projection (UMAP)[55] (Supplementary Fig. 3b). Next, the dataset was split into a tumor and a TME dataset (Supplementary Fig. 3c, d). The cell type assignment of the TME was done based on a Cite-seq BM reference dataset[56] by using the multimodal reference mapping approach from Stuart et al.[22] In brief, the cell type assignment of each sample works as follows: anchors were defined between the reference and each query sample and then each sample was individually mapped to the reference. In a next step, all annotated samples were merged as they have been integrated into a common reference space and then visualized. The tumor cells within the CD138-negative fraction, respectively the TME cells from the CD138-positive fraction were removed from downstream analysis. The separate tumor and TME

datasets were then again normalized with SCtransform[53]. Cell-cycle scores were calculated with the Seurat function "CellCycleScoring".

Differential expression analysis was performed with the standard Seurat function Find(All)Markers (parameters: min.pct = 0.25, logfc.-threshold = 0 for subclone comparison between focal lesions and RBM, respectively logfc.threshold = 0.25 for the general subclone comparisons). Pearson correlations between subclones were calculated based on the genes, which were also used for the differential expression analysis (min.pct = 0.25, logfc.threshold = 0). For gene set enrichment analysis the hypeR R-package (v1.6.0)[57] was used with the MSigDB HALLMARK geneset[58].

Gene expression signatures were calculated with the Seurat function AddModule Score (https://satijalab.org/seurat/index.html) by using recently published signatures to determine the level of exhaustion (*TIGIT*, *HAVCR2*, *CTLA4*, *PDCD1*, *LAG3*, *LAYN*) and cytotoxicity (*NKG7*, *CCL4*, *CST7*, *PRF1*, *GZMA*, *GZMB*, *IFNG*, *CCL3*) of T-cells[28].

VDJ filtered_contig_annotations files were loaded into R, combined from all patients (separately for TCR and BCR) and mapped to the Seurat object by using the scRepertoire package (v0.99.15)[59]. Clonal T-cells, which made up ≥1% of the total T-cell population and showed at least 5 cells in one of the paired samples, were called expanded. Clonal T-cells ≥5% of all T-cells were called hyper-expanded. T-cells with only one successfully sequenced chain were merged with the corresponding clonal T-cells with complete chain information. To test for antigen specificity, the CDR3 sequences of the expanded TCRs were looked up in the curated databases VDJdb[29] and McPas-TCR[30].

To call mutations in scRNA-seq data, SNV positions from the WGS data (hg19) were transformed to hg38 using rtracklayer (v1.50.0)[60] and mutation calling in the scRNA-seq data was performed using Vartrix with default parameters (https://github.com/10XGenomics/vartrix). This mutation information was then mapped to the Seurat tumor objects.

## Preprocessing and analysis of single-cell ATAC-sequencing data

Raw scATAC-seq reads were aligned to the reference genome GRCh38 using CellRanger ATAC (10X Genomics, version 1.2). For downstream analysis, the fragment files of all samples (4x paired focal lesion and random bone marrow specimens) were loaded into the ArchR framework (v1.0.2)[61] with default parameters including doublet detection. In addition, based on the quality control plots all cells with a transcription start site (TSS) enrichment score <8, <3000 fragments, a doublet enrichment score >6 and a predicted doublet score >200 were filtered out (Supplementary Fig. 3e–g, Supplementary Table 8). This resulted in 7741 cells with a median TSS of 13.726 and median fragment size of 18,239. First, all cells from the different patients together were normalized using iterative latent semantic indexing (LSI)[62,63] and clustered with a resolution of 0.5 (Supplementary Fig. 3h). Gene activity imputation was performed using *MAGIC*[64] and peak calling with *MACS2* (extendSummits = 750)[65]. Plasma/myeloma cells could be distinguished from other immune cells based on the gene activity score of CD138/SDC1. One cluster containing plasma cells from different patients was defined as a normal plasma-cell reference for CNA-calling analogous to scRNA-seq (Supplementary Fig. 3h).

Motif deviations were calculated based on the JASPAR database[66]. For pseudobulk TF motif analysis, the MotifMatrix was extracted from the ArchR-project using the "getMatrixFromProject" function. For each TF motif, a mean value was calculated over all cells from the corresponding sample. The most highly variable TFs were visualized using pheatmap in R. Co-accessibility between genomic regions was separately calculated for timepoints and CNA subclones, respectively, adjusting the ArchR framework to single-cell resolution without aggregation of cells. The degree of co-accessibility in the background was determined by randomly shuffling the accessibility values over cells and peaks as described previously[67]. The 99th percentile of the

maximum shuffled background co-accessibility score was used as a threshold to determine true co-accessible links. Co-accessible links were further evaluated by percent of accessible cells in the linked peak pairs[68].

### Identification of subclones in single-cell sequencing data

For scRNA-seq, subclones were identified based on the presence of subclonal CNAs. CNAs were predicted using the R-package InferCNV (v1.6.0)[13]. The prediction is based on expression averages of adjacent genes over large genomic regions. As reference we used normal plasma cells from patients in our dataset, which clustered together (Supplementary Fig. 3d). As input for InferCNV we used normalized counts of the patients together with a gene ordering file built based on the 10X Genomics reference file for running CellRanger. Importantly, we only included regions with subclonal CNAs according to WGS. The cells were hierarchically clustered (method ward.D2) based on the InferCNV output and the clustered dendrogram was cut according to visually identified subclones using the R-package dendextend (v1.15.2)[69]. For scATAC-seq, CNAs were called using a script published by Lareau et al.[14] (https://github.com/caleblareau/mtscATACpaper_reproducibility/tree/master/cnv_compute). Briefly, overlapping 10 Mb genome-wide bins were constructed from the fragments file. For all cells passing QC criteria a bin by cell matrix was computed separately for the malignant and the normal plasma cells from all patients (Supplementary Fig. 3h), and a z-score matrix was calculated using the normal plasma cells as reference. Next, similar to the scRNA-seq approach, we performed hierarchical clustering of regions with subclonal CNAs according to WGS. Heatmaps of clustered z-scores for chromosomes from single cells were visually inspected and compared to scRNA-seq heatmaps and WGS chromosomal profiles to identify congruities and conflicting results. The location of chromosomal regions, which were used for clustering, and the number of consecutive rounds, in case an iterative process was required, are shown in Supplementary Table 9. The recently published script[70] for the WGS-guided subclone identification in scRNA-seq and scATAC-seq data is available at: https://github.com/a-poos/MM_subclones.

### Immunohistochemistry

Representative tissue blocks containing MUM1-positive myeloma cells were selected. The blocks were sectioned with a standard microtome at 2 μm thickness. Subsequently, the slides were dried overnight at room temperature. Immunohistochemical staining for MUM1, CD68, CD4, CD8, CD138, Ki67, CXCL12, CXCL7 and CXCR4 was performed using the automated immunostainer Ventana Benchmark Ultra (Roche, USA). Used antibodies are shown in Supplementary Table 6. For image acquisition of stained slides the Aperio AT2 slide scanner at 40x magnification and the manufacturer's acquisition software suite were used (Leica Biosystems, Nussloch, Germany).

Images were analyzed using the QuPath software (v0.3.2). Therefore, the images were imported using the Bioformats builder. Detection of all cells and the positive stained cells within the specimen was done using the integrated "positive cell detection" module of QuPath with "Hematoxylin" as the reference channel. The module estimates the full extent of each cell based upon a constrained expansion of the nucleus region and calculates up to 66 measurements of intensity and morphology. For the quantification of CD4-, CD8- or CD68-positive cells, the threshold for positive detection was calculated for each image analyzing at least 30 CD4, CD8 and CD68 cells for their DAB staining signal and the mean value was then formed and used. To correlate the proportion of these cells with the tumor load within a slide, the level of plasma cell infiltration was visually scored in each slide by an expert pathologist from 1 to 4, representing 5–25%, 26–50%, 51–75% and 76–100% infiltration, respectively. For the

quantification of CXCL12 a three-tiered threshold was used, which was calculated by analyzing at least 30 CXCL12 positive plasma cells, which were identified based on their morphology, of 10 randomly selected whole slide images for their DAB staining signal. The values were set to 0.2 (1+), 0.4 (2+) and 0.8 (3+).

To determine the Ki-67 index of myeloma cells, slides co-stained for CD138 (DAB) and Ki-67 (FastRed) were analyzed. First, a "single measurement classifier" for detecting CD138-positive cells was created. For classification the following settings were used: object filter: "cells", channel filter: "DAB", measurement: "cell:DAB mean". The threshold for positive detection was calculated for each image analyzing at least 30 myeloma cells for their CD138 bright field signal and the mean value was then formed. Cells above the threshold were classified as "myeloma cells" and cells below the threshold were classified as "hematopoiesis". Next, the KI-67 index was analyzed within the myeloma cells using the internal "cell intensity classification" module of QuPath. The threshold was calculated as described for the CD138 immunostaining. A final check of all stained and detected cells was performed by an expert pathologist.

### Enzyme-linked immunosorbent assay

For ELISA, either paired dry pellets or viably frozen cells were used. Viably frozen CD138+ BMMCs were thawed and washed twice with 1x PBS. Cells were lysed using ice-cold lysis buffer ABIN0-007-3 (antibodies-online GmbH, Aachen, Germany). After 30 min on ice, the tubes were subjected to ultrasonication for 3x 15 s. Protein concentrations were determined using the Bicinchoninic acid reaction (Pierce™ BCA Protein Assay Kit, ThermoFisherScientific, Waltham, Massachusetts, USA) according to the manufacturer's instructions. CXCL7 and CXCL12 were quantified using the Pro-Platelet Basic Protein (Chemokine (C-X-C Motif) Ligand 7) (PPBP) and Chemokine (C-X-C Motif) Ligand 12 (CXCL12) ELISA kits offered by antibodies-online, respectively. Due to limited material, we used between 750-1500 ng total protein per reaction, with matched protein loads for paired samples. Quantifications were performed twice in duplicates.

### ChIP-seq data

Publicly available IRF4 ChIP-seq data[20] for MM cell line KMS12BM were downloaded from the European Nucleotide Archive (accession number: PRJEB25605; https://www.ncbi.nlm.nih.gov/bioproject/PRJEB25605/) and peaks were called using the nfcore/chipseq pipeline[71] (v.1.0.0) with default settings (narrowPeaks).

### Statistical methods

Statistical analyses were carried out using the R software package v4.0.2. Group comparisons of continuous variables were done using the two-sided Wilcoxon rank sum test for unpaired samples and the two-sided Wilcoxon signed rank test for paired samples. Correlation coefficients were determined using Pearson correlation. The Kaplan-Meier method was used for survival analyses. Overall survival was time from enrollment to death of any cause. Cox proportional hazards regression was used to estimate hazard ratios and 95% confidence intervals.

### Reporting summary

Further information on research design is available in the Nature Portfolio Reporting Summary linked to this article.

## Data availability

WGS, RNA, scRNA, scATAC, TCR and BCR sequencing data of this study have been deposited at the European Genome-phenome Archive with the study identifier EGAS00001006090 and are available on request from the associated Data Access Committee (hipo_daco@dkfz-heidelberg.de) due to them containing patient information under

controlled access. Access will be granted to commercial and non-commercial parties according to patient consent forms and data transfer agreements. We have an institutional process in place to deal with requests for data transfer and aim for a rapid response time. The duration of data access after approval is limited to 36 months.

The publicly available gene expression profiling datasets used in this study are available at ArrayExpress under accession code E-MTAB-2299 and Gene Expression Omnibus under accession code GSE19784 [72]. The publicly available whole exome sequencing data used in this study are available from the European Genome-phenome Archive under accession code EGAS00001002111 [5]. IRF4 ChIP-seq data of the MM cell line KMS12BM used in this study are publicly available from the European Nucleotide Archive with the accession number PRJEB25605 [20]. The remaining data are available within the Article, Supplementary Information or Source Data file. Source data are provided with this paper.

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

## Acknowledgements

L.J. was supported by a Heidelberg School of Oncology (HSO2) fellowship from the National Center for Tumor Diseases (NCT) Heidelberg and by the International Myeloma Working Group. This project was supported by the Black Swan Research Initiative, the Dietmar-Hopp Foundation and DKFZ-Heidelberg Center for Personalized Oncology (DKFZ-HIPO) funding. For the publication fee we acknowledge financial support by Deutsche Forschungsgemeinschaft within the funding programme "Open Access Publikationskosten" as well as by Heidelberg University. The authors acknowledge support by the state of Baden-Württemberg through bwHPC and the German Research Foundation (DFG) through grant INST 35/1134-1 FUGG. The authors gratefully acknowledge the data storage service SDS@hd supported by the Ministry of Science, Research and the Arts Baden-Württemberg (MWK) and the German Research Foundation (DFG) through grants INST 35/1314-1 FUGG and INST 35/1503-1 FUGG. We thank the Sample Processing Lab (SPL), the High Throughput Sequencing unit of the Genomics & Proteomics Core Facility and the Omics IT and Data Management Core Facility (ODCF) of the German Cancer Research Center (DKFZ) for providing excellent services. We would also like to thank all patients who participated in this study.

## Author contributions

A.D-S., C.Sa., and U.H. performed functional imaging and analysis thereof; B.B., B.W., C.L., C.M-T., C.R., C.Sc., F.V.R., H.G., J.H., L.J., M-S.R., M.Z., S.He. and S.Sa. provided study material and patient samples; A.Ba., S.Hu., P.R. and S.Q. processed samples; A.Br., G.M., J.-P.M., K.B., N.P., R.L., D.V., S.Sc., S.T. and K.R. performed experimental work; A.Br., A.P., C.Sc., L.S.-B., S.Ha., S.St., M.P., L.J., L.R., and N.W. analyzed data; A.P., L.J. and N.W. wrote the manuscript; all authors revised and approved the manuscript; N.W. designed the study; S.Sa. and N.W. supervised the study.

## Funding

## Competing interests

The authors declare no competing interests.

## Additional information

[1]Department of Internal Medicine V, Heidelberg University Hospital, Heidelberg, Germany. [2]Clinical Cooperation Unit Molecular Hematology/Oncology, German Cancer Research Center (DKFZ), Heidelberg, Germany. [3]Department of Pathology, Heidelberg University Hospital, Heidelberg, Germany. [4]Myeloma Center, University of Arkansas for Medical Sciences, Little Rock, AR, USA. [5]Heidelberg Institute for Stem Cell Technology and Experimental Medicine (HI-STEM gGmbH), Heidelberg, Germany. [6]Division of Stem Cells and Cancer, German Cancer Research Center (DKFZ) and DKFZ-ZMBH Alliance, Heidelberg, Germany. [7]Division of Chromatin Networks, German Cancer Research Center (DKFZ) and BioQuant, Heidelberg, Germany. [8]Single Cell Open Lab, German Cancer Research Center (DKFZ) and BioQuant, Heidelberg, Germany. [9]Institute of Health (BIH) at Charité—Universitätsmedizin Berlin, Berlin, Germany. [10]Berlin Institute for Medical Systems Biology, Max Delbrück Center for Molecular Medicine in the Helmholtz Association, Berlin, Germany. [11]Department of Hematology, Oncology and Tumor Immunology, Charité University Medicine, Berlin, Germany. [12]Division Computational Genomics and Systems Genetics, German Cancer Research Center (DKFZ), Heidelberg, Germany. [13]Wellcome Sanger Institute, Wellcome Trust Genome Campus, Cambridge, UK. [14]Department of Orthopedic Surgery, Heidelberg University Hospital, Heidelberg, Germany. [15]Department of Radiology, Heidelberg University Hospital, Heidelberg, Germany. [16]Department of Nuclear Medicine, University Hospital Heidelberg, Heidelberg, Germany. [17]Clinical Cooperation Unit Nuclear Medicine, German Cancer Research Center (DKFZ), Heidelberg, Germany. [18]National Center for Tumor Diseases (NCT), Heidelberg, Germany. [19]Department of Medicine, Roswell Park Comprehensive Cancer Center, Buffalo, NY, USA. [20]Department of Internal Medicine 2, University Hospital of Würzburg, Würzburg, Germany. [21]Mildred Scheel Early Career Center (MSNZ), University Hospital of Würzburg, Würzburg, Germany. [22]These authors contributed equally: Lukas John, Alexandra M. Poos. [23]These authors jointly supervised this work: Sandra Sauer, Niels Weinhold. ✉e-mail: niels.weinhold@med.uni-heidelberg.de

