## [Peer Review File · Nature Communications]

Resolving the Spatial Architecture of Myeloma and its Microenvironment at the Single-Cell LevelReviewers' Comments:

Reviewer #1:

Remarks to the Author:

This is a very interesting manuscript interrogating spatial heterogeneity of focal lesions versus bone marrow myeloma cells from newly diagnosed patients at the subclonal level, comparing bulk and single cell transcriptomes. Unique features of the manuscript include the spatial heterogeneity of the tumor microenvironment cells including the Macrophage and T cells. The manuscript is well-written, clear and thought-provoking particularly as they highlighted the contrasting results of downregulation of MM CXCL7 and CXCL12 and decreased macrophage infiltration in the focal lesions. The downregulation of MHC I and II in the focal lesions also support immune escape theory. The impact of this work is the better understanding of disease evolution and, if confirmed in larger studies or in serial samples from the same patients over time, may impact future treatment.

My comments are below. Most are minor

1. Since the number of focal lesions of already been associated with prognosis, and the authors hypothesized that the lesion size may explain the contradicting gene signature in their study compared to prior report, please provide detail in each patient the number, size and locations of lytic lesions. In addition, please provide information on how lesions are chosen for biopsy if more than one is present in a patient. Also, to aid in the understanding of the
2. Sample processing could affect gene expression and the # of clones detected. Please provide workflow on how biopsies are done (are the biopsies of focal lesions and bone marrow done concurrently?) and sample processing
3. Three paramedullary samples were included, but biologically it is not clear whether these remain marrow dependent, similar to the intramedullary disease or leading towards EMD. Are there any difference in the spatial heterogeneity of the MM cells and TME in these 3 cases, compared to the remaining 12?
4. In table 1, please clarify whether FISH risk (-) means negative for standard FISH panel or data not available. A quarter of cases have chromosome 1 abnormality, which predict for high risk disease, was there any spatial differences between these cases and the normal-FISH group?
5. The inverse correlation of macrophage infiltration and present plasma cells is intriguing. In the histology sections, the % macrophages are variable in different areas with the lowest number present in areas with the heaviest infiltration. The authors postulated that downregulated MM CXCL7 and CXCL12 may explain this. To go beyond the hypothesis, please consider staining the histology section for CXCL7 and CXCL12 in the tumor cells. Also correlation between the T-cell clones and numbers vs. macrophage could further be interrogated for correlation.
6. Gene expression may be affected by cell cycle. Please add info on the % plasma cells and Ki67% between the focal lesions and BM. If they are different, please discuss further how that may impact the data interpretation.

Reviewer #2:

Remarks to the Author:

John and Poos et al uncover features of spatial heterogeneity in the subclonal architecture and molecular signatures in both tumor and microenvironment using paired specimens from 15 newly diagnosed MM patients. They show that spatial tumor heterogeneity is a frequent event in MM patients with focal lesions. Using a comprehensive approach, they discovered key features in spatial heterogeneity including down-regulation of chemoattractant cytokines and depletion of myeloid lineage cells in focal lesions. These findings can provide some insights in the processes underlying increasing independence of MM cells from the BM microenvironment.

The studies are very interesting and novel but preliminary given the number of samples.

A few comments are included below:

1. The number of samples is too small to make strong statements especially that not all samples had all the multiomic analyses described.
2. The authors need to differentiate NA and WT in mutation calling in single-cell. Detection power should help in differentiating NA and WT. Is the fraction mutant positive expressed over WT or over WT+NA?
3. Related to Figure 2, are there any shared transcriptional differences (signature) observed between site-unique subclones and other disseminated clones? If any, this could provide some insights into "myeloma metastasis" and/or focal formation.
4. It's difficult to understand the story on interferon and HLA. It seems that the difference in transcriptional signature observed between these two CNA-defined subclones could reflect the transcriptional plasticity induced by the microenvironment. Did you see any epigenomic mechanism suggested by your scATAC-seq data for these subclones?
5. It would be great to examine the protein expression level of CXCR7/CXCL12. DESeq2 should incorporate sample purity as a covariate at the very least.
6. CXCR4 in myeloma cells and their spatial distribution within the tissue in focal lesion compared to random bone marrow also needs to be confirmed at protein level. CXCR4 is internalized in the marrow and RNA levels may not reflect surface protein expression.
7. Related to Figure 5b, this observation of decrease in monocyte/macrophage lineage cells was compatible with any other data derived from other experimental approaches like flow cytometry or cell counting from a smear sample?
8. We need the details of linear regression to explain variation in macrophages L215–218. If the predictors are correlated, what is the variance inflation factor? What is the expected macrophage diminution factor given plasma cell increase vs observed in IHC?
9. Do you have any serial samples from these patients, with progression from MGUS or SMM to the current NDMM or even after treatment, this would be really interesting?
10. In general the manuscript is missing confidence intervals of estimates (CCF, TCR clonality, proportion of macrophages, etc) and to Cis overlap for example between BM and FL). Also distributions should be described with IQR or standard deviation or 95% central interval in addition to mean/median.

Minor comments:

1. Check the clone names used in IHC studies. The clone names for antibodies of CD4, CD8 and CD68 are the same one 'MIB-1'.
2. Related to Supple Table 7, there is no T cell subtype detail yet the legend mentions it.

Reviewer #3:

Remarks to the Author:

In this study, John et al. attempted to decipher spatial heterogeneity of multiple myeloma (MM) and its microenvironment between random bone marrow (RBM) samples from the iliac crest and paired focal lesion (FL) specimens at single-cell level. The manuscript is well written and documented with informative figures. This study generated a large amount of data, and the huge amount efforts spent

for this work need to be recognized. While the experimental and bioinformatical effort invested into this paper is remarkable, its results are not truly novel as many of them have been described in previous studies (to which the authors also refer to). Overall, the authors attempted to elucidate a detailed description on subclonal architecture, expression signatures and the composition of the microenvironment in FL, which are quite descriptive. Furthermore, the major conclusions in this study are drawn from a small sample size (only 5 or 6 patients for single-cell sequencing) and lack further validation. Many claims are speculative and are not tested, nor based on independent cohorts neither experimentally. Many further studies are required to validate the data and understand the interactions within immune cell populations (e.g., the depleted macrophages and skewed T-cell repertoire in FL) and between the host immune system and the tumor. Furthermore, there is little novel insight from the study, and the low number of patients undermined all conclusions. This paper is more suited for a more specialized journal.

Specific comments:

1. The authors claims that the genetically identical subclones at different BM sites have different gene expression signatures, which is contributed by transcriptional plasticity. This conclusion is one of the main findings of this study, but it is very obvious indeed. Of course, spatially different TME has different gene expression signatures. What is the underlying molecular mechanism? The authors did not elaborate it adequately.
2. To explain why macrophages are depleted in focal lesions, the authors correlated the proportion of macrophages with the level of plasma cell infiltration. However, this result seems not convincing. In fig5c, there is almost no extra space for other cells when the infiltration of plasma cells is greater than 76%, thus the decreased proportion of macrophages or other cells is as expected/not surprising. Besides, in Suppl. Table 8, the proportions of plasma cells in RBM and FL pair do not show great difference. Could the authors provide the mechanism explanation why macrophages are depleted in intramedullary focal lesions?
3. The analysis of scATAC-seq data is superficial. The data were used only for CNA-calls. Is there other useful information that could be drawn from scATAC-seq? For example, the integration of the RNA and ATAC-seq data would benefit the molecular mechanism analysis.

Minor concerns:

1. One of the main findings that chemoattractant cytokines CXCL7 and CXCL12 are downregulated in focal lesions, should be validated on the protein level.
2. Line 123, the name of "clone 6B" is abrupt, which should be replaced by "subclone 6b" as the same in Fig.2C.
3. Line 463, there is an incomplete expression: "multimodal reference mapping approach from".
4. The order of group color should keep consistent in Fig.1c.
5. The color of Fig. 2b is too miscellaneous.

Revision of manuscript # NCOMMS-22-13015-T

REVIEWER COMMENTS

Reviewer#1

This is a very interesting manuscript interrogating spatial heterogeneity of focal lesions versus bone marrow myeloma cells from newly diagnosed patients at the subclonal level, comparing bulk and single cell transcriptomes. Unique features of the manuscript include the spatial heterogeneity of the tumor microenvironment cells including the Macrophage and T cells. The manuscript is well-written, clear and thought-provoking particularly as they highlighted the contrasting results of downregulation of MM CXCL7 and CXCL12 and decreased macrophage infiltration in the focal lesions. The downregulation of MHC I and II in the focal lesions also support immune escape theory. The impact of this work is the better understanding of disease evolution and, if confirmed in larger studies or in serial samples from the same patients over time, may impact future treatment. My comments are below. Most are minor.

1. Since the number of focal lesions has already been associated with prognosis, and the authors hypothesized that the lesion size may explain the contradicting gene signature in their study compared to prior report, please provide detail in each patient the number, size and locations of lytic lesions. In addition, please provide information on how lesions are chosen for biopsy if more than one is present in a patient. Also, to aid in the understanding of the

Response: We would like to thank the reviewer for the positive feedback and the constructive critique! As requested, we now provide for each patient the number, size and location of lytic lesions in the new Suppl. Table 11. For CT-guided biopsies, focal lesions were chosen according to accessibility and minimal risk for the patient. In case of multiple eligible lesions, the best accessible lesion was selected. We have added this information to the methods paragraph: **"Focal lesions for CT-guided biopsies were chosen according to accessibility and minimal risk for the patient. In case of multiple eligible lesions, the best accessible lesion was selected. CT-guided biopsies were performed in a separate session within one week after collection of the diagnostic sample from the iliac crest. All biopsies were performed prior to initiation of treatment. For each biopsy site 10-15 ml of BM were collected and processed on the same day."** (page 15, lines 463-467)

2. Sample processing could affect gene expression and the # of clones detected. Please provide workflow on how biopsies are done (are the biopsies of focal lesions and bone marrow done concurrently?) and sample processing.

Response: We would like to thank the reviewer for pointing to this potential confounder. CT-guided biopsies from intramedullary focal lesions were performed in a separate session within one week after collection of the diagnostic sample from the iliac crest, and all samples were collected prior to initiation of treatment. Aspirates from intramedullary lesions and random bone marrow sites were processed using the same methodologies. From each site 10-15 ml of bone marrow were collected. Cell sorting using immuno-magnetic CD138-positive selection was performed on the same day of sampling.

For paramedullary lesions, we used a different procedure for sample processing, since aspirates are not feasible for solid tissue. For patient P09, we isolated DNA for whole-genome sequencing from a fresh frozen surgically resected specimen with myeloma cell infiltration of 84%. For patients P12, P13, P16, and P17 samples from surgical resections of paramedullary lesions were minced and enzymatically digested for 15 min at 37°C in MEM alpha Medium (Gibco, USA) containing 1 mg/ml Collagenase II, 0.8mg/ml Dispase (both Gibco, USA) and 0.1mg/ml DNase I (SigmaAldrich, USA). Cells were released by pipetting repetitively (30x). The reaction was stopped using a quenching buffer (Calcium-free PBS + 5mM EDTA + 2% FCS). For sequencing, MM cells were sorted (CD38 high, CD45RA-) using a FACS Aria (BD Biosciences). Dead cells were identified and excluded using eFluor-506 (ThermoFisherScientific, USA). Sorted cells were stored in RLT buffer (Qiagen, Hilden, Germany) before bulk sequencing. In the revised manuscript the type of processing is shown for each sample in the new Suppl. Table 12:

PID	Random sample			Focal Lesion		
	Sample type	Sorting method	Purity (WGS)*	Sample type	Sorting method	Purity (WGS)*
P01	Aspirate	Robosep	0.9	Aspirate	Robosep	0.8
P02	Aspirate	Robosep	0.96	Aspirate	Robosep	0.92
P03	Aspirate	Robosep	1	Aspirate	Robosep	1
P04	Aspirate	Robosep	0.92	Aspirate	Robosep	0.98
P05	Aspirate	Robosep	0.78	Aspirate	Robosep	0.8
P06	Aspirate	Robosep	NA	Aspirate	Robosep	NA
P07	Aspirate	Robosep	0.98	Aspirate	Robosep	0.86
P08	Aspirate	Robosep	0.9	Aspirate	Robosep	0.84
P09	Aspirate	Robosep	0.96	Surgical resection	fresh-frozen tissue, no sort	0.84
P10	Aspirate	Robosep	0.82	Aspirate	Robosep	0.92
P11	Aspirate	Robosep	0.52	Aspirate	Robosep	0.9
P12	Aspirate	Robosep	0.98	Surgical resection	FACS Aria	0.98
P13	Aspirate	Robosep	0.88	Surgical resection	FACS Aria	0.98

P14	Aspirate	Robosep	0.98	Aspirate	Robosep	0.91
P15	Aspirate	Robosep	1	Aspirate	Robosep	1
P16	Aspirate	Robosep	0.86	Surgical resection	FACSAria	0.98
P17	Aspirate	Robosep	0.7	Surgical resection	FACSAria	0.98

* The tumor purity was estimated based on histograms for the variant allele frequency (VAF) of mutations according to whole genome sequencing.

We would like to apologize that this information was missing in the first version of the manuscript. Hence, for single-cell analyses and bulk RNA-seq of paired intramedullary lesions and random bone marrow aspirates, sample processing, as well as single-cell and bulk sequencing were done using the same methodologies. Furthermore, subclones were detected using a dual approach combining whole-genome-sequencing and single-cell sequencing. Only subclones were counted that were detected by both methods. Yet, subclones could differ in their dependency on the bone marrow microenvironment, and highly dependent subclones could strongly respond to sample processing. We now discuss this issue in the Discussion section (page 13, lines 412-413).

As mentioned, for paramedullary lesions, the procedure for myeloma cell enrichment was different involving tissue disintegration steps. Thus, we cannot exclude that some differences in expression profiles between paired paramedullary/iliac crest samples were due to sample processing. However, for the top differentially expressed genes (*CXCL7*, *CXCL12*, *ADM*, and *MYLIP*), the same direction of effect was seen in intramedullary/iliac crest and paramedullary/iliac crest pairs from our institute as well as in the expression data for 250 patients from the University of Arkansas, which were based on identically processed paired samples. Therefore, we are confident that differential expression of these four genes was not due to differences in sample processing.

3. Three paramedullary samples were included, but biologically it is not clear whether these remain marrow dependent, similar to the intramedullary disease or leading towards EMD. Are there any differences in the spatial heterogeneity of the MM cells and TME in these 3 cases, compared to the remaining 12?

Response: We would like to thank the reviewer for this suggestion! To check if there are differences between intramedullary and paramedullary lesions, we increased the sample size by two more patients (P16, P17) with paramedullary lesions. We observed a trend towards more heterogeneous (=unshared+enriched) mutations in patients with paramedullary disease compared to patients with intramedullary lesions (mean 33.4% (range: 9.65-58.83%) vs. 14.6% (1.02-41.08%), $p=0.07$). However, despite lower numbers the majority ($n=8/11$) of patients with intramedullary lesions showed spatial heterogeneity. This result is shown in the new Extended Data Fig. 2a.

Extended Data Fig. 2a: Proportion of heterogeneous mutations (unshared+enriched) in intra- vs. paramedullary samples. The p-value was calculated using the two-sided Wilcoxon rank sum test.

The downregulation of *CXCL12* at the mRNA level was slightly more pronounced in paramedullary lesions (mean log₂ fold change=4.16, range=1.01-7.56), but it was also seen in intramedullary lesions (mean log₂-fold =1.63, range=0.76-2.89). For the other three genes (*CXCL7*, *ADM*, and *MYLIP*) we also did not observe significant differences between the two types of lesions. These new results are shown in the new Extended Data Fig. 5a.

Extended Data Fig. 5a: Line plots for the log₂-normalized bulk RNA-seq expression values of the two down-regulated genes *CXCL7* and *CXCL12* as well as the two upregulated genes *ADM* and *MYLIP* are shown for intra- and paramedullary lesions and the paired random bone marrow samples.

Next, we directly compared intra- to paramedullary lesions and found a total of 666 genes, which were differentially expressed between them. These genes showed an enrichment for MTORC signaling, interferon alpha/gamma response as well as TNF alpha signaling via NFKB. We do not present this analysis in the manuscript, since we cannot exclude that differences in sample processing resulted in expression differences (please see our response to your comment #2). Yet, we now discuss that future analyses should longitudinally compare the expression profiles of intra-, para- and extramedullary lesions to further decipher the changes that lead towards bone-marrow independence and EMD (page 12, lines 386-394).

4. In table 1, please clarify whether FISH risk (-) means negative for standard FISH panel or data not available. A quarter of cases have chromosome 1 abnormality, which predict for high

risk disease. Was there any spatial differences between these cases and the normal-FISH group?

Response: Thank you for outlining this potential source of misunderstanding! In table 1, which is now Suppl. Table 10, FISH risk (-) meant absence of t(4;14), t(14;16), gain(1q21), and del(17p13). In the revised manuscript, we use “standard” and “high risk” with “high risk” being defined as presence of t(4;14), t(14;16), gain(1q21), and/or del(17p13). Comparing the four gain(1q) cases with 12 cases without this aberration, there was no statistically significant differences in the proportion of heterogeneous mutations (gain(1q): mean=22.79%, range=2.84-38.75%, w/o gain(1q): mean=19.70%, range=1.02-58.83%, $p=0.68$). Next, to account for the small sample size in this study, we analyzed data from a recently published multiregion whole-exome-sequencing study (27 newly diagnosed patients w/o & 15 with gain(1q)). There were also no significant differences in the proportion of heterogeneous mutations between the two groups (gain(1q): mean=18.47%, range:0-43.86%, w/o gain(1q): mean=24.44%, range:0-78.58%, $p=0.92$). The new results are shown in the new Extended Data Fig. 2b-c.

Extended Data Fig. 2b-c: Frequency of heterogeneous mutations (unshared+enriched) in patients with gain 1q vs. patients with no gain 1q in the Heidelberg (left panel) and the UAMS patient cohort (right panel). The p-value was calculated using the two-sided Wilcoxon rank sum test.

5. *The inverse correlation of macrophage infiltration and present plasma cells is intriguing. In the histology sections, the % macrophages are variable in different areas with the lowest number present in areas with the heaviest infiltration. The authors postulated that downregulated MM CXCL7 and CXCL12 may explain this. To go beyond the hypothesis, please consider staining the histology section for CXCL7 and CXCL12 in the tumor cells. Also correlation between the T-cell clones and numbers vs. macrophage could further be interrogated for correlation.*

Response: We thank the reviewer for the suggestion to correlate T-cells with the proportion of macrophages. We found a strong positive correlation (Pearson correlation 0.92) between the number of expanded T-cell clones and the proportion of CD8+T-cells, while the number of expanded T-cell clones was negatively correlated with the proportion of CD4-positive T-cells

(Pearson correlation -0.52) and macrophages (Pearson correlation -0.40). These results are now shown in Extended Data Fig. 8e.

Extended Data Fig. 8e: Pearson correlation between the number of expanded T-cell clones (x-axis) and the proportion of CD8-positive T-cells (left), CD4-positive T-cells (middle) and macrophages (right) for the 6 patients with scRNA-seq data. Paired samples were plotted separately and colored by patient. The linear regression line is depicted in black, where the grey area marks the 95% confidence interval.

We also thank the reviewer for the excellent suggestion to stain histology sections! Unfortunately, myeloma cells were negative in histology sections stained for CXCL7, while megakaryocytes were clearly positive, suggesting that CXCL7 protein levels were below the detection limit in myeloma cells. In line, CXCL7 protein expression in CD138-sorted myeloma cells was at or even below the detection limit of ELISA.

Example of a bone marrow histology section from a myeloma patient stained for CXCL7 (magnification in the right panel).

For CXCL12, we surprisingly did not observe consistent differences in the proportion of CXCL12-positive MM cells in focal lesions compared to random bone marrow specimens (median: 5 (range:0.04-20.6) vs. 1.35 (0.044-20.1), $p=0.64$ in two-sided Wilcoxon signed rank test).

In the revised version of the manuscript, we show paired samples from patient P13 as an example. The paramedullary lesion of this patient presented with a 215-fold down-regulation of the *CXCL12* gene compared to the random bone marrow sample but was among the

samples with the highest proportion of CXCL12-positive myeloma cells and showed similar results in ELISA compared to the paired random bone marrow sample. In this patient, both CXCL12-negative and -positive myeloma were seen in regions with the highest infiltration. These results are presented in the new Fig. 3d and Extended Data Fig. 5.

Extended Data Fig. 5b (left) and Fig. 3d (right). In the extended data figure, the ratios for CXCL12 protein expression in CD138-enriched myeloma cells in paired random bone marrow (RBM) and focal lesion (FL) samples from 6 patients are shown. Expression was quantified in two experiments with duplicates using ELISA. The dotted line marks a ratio of 1, corresponding to no differences between paired samples. The p-value was calculated using a linear mixed-effects model. In **Fig. 3d** the proportion of CXCL12-positive myeloma according to IHC is shown. In the left panel paired tissue slides from patient P13, which were stained for CXCL12, are depicted as examples. The line plot in the right panel shows the proportion of CXCL12-positive myeloma cells in paired samples from 8 patients, which were selected based on availability of bulk RNAseq. The p-value was calculated using the Wilcoxon signed-rank-test.

Given that downregulation of CXCL12 expression at the mRNA level was confirmed in a large independent cohort of 250 patients with paired samples, we don't believe that changes in gene expression profiles were simply artefacts. We rather think that the difference between mRNA and protein levels was due to a compensation mechanism. One possible mechanism could be increased internalization of CXCL12 by myeloma cells. In line, patient P13, who presented with the highest difference in CXCL12 RNA expression between focal lesion and bone marrow, showed a 5-fold expression upregulation of CXCR7 in the lesion. However, this was seen in just one patient. Alternatively, the supply of myeloma cells with CXCL12 could be increased in focal lesions. Supporting this assumption, we observed a significant increase in mesenchymal stromal cells, a major source of CXCL12 in the bone marrow, in focal lesions in 8 patients with paired samples. Please see our response to comment #5 of reviewer #2 for more details. This finding suggests a compensation mechanism mediated by the microenvironment which would be intriguing to examine in further studies.

Yet, as further research is required to explain the difference between mRNA and protein levels and since we cannot conclude if changes in CXCL12 and the depletion of macrophages are the cause or the result of the expansion of advanced subclones in focal lesions, we removed the statement from the Discussion that down-regulated CXCL7 and CXCL12 could explain the depletion of macrophages in regions with a high plasma cell infiltration.

6. Gene expression may be affected by cell cycle. Please add info on the % plasma cells and Ki67% between the focal lesions and BM. If they are different, please discuss further how that may impact the data interpretation.

Response: As requested, we have co-stained CD138 and Ki67 in core biopsies of 8 patients with bulk RNAseq to address the potential link between proliferation and expression differences. The proportion of Ki67-positive MM cells was not significantly different between paired focal lesion and random bone marrow (focal lesion median: 5.7% (0.2-48.6%) vs. RBM median: 9.2% (2.7-16.7%), $p=0.84$). The results are shown in the new Fig. 3c. We conclude that consistent differences in expression profiles are not due to differences in proliferation.

Fig. 3c: Ki-67 index of myeloma cells. Slides were co-stained for CD138 (DAB) and Ki-67 (FastRed). The random bone marrow (RBM) and the focal lesion (FL) of patient P13 are shown as examples in the left panel. In the right panel a line plot is shown for the proportion of Ki67-positive plasma cells in paired samples from 8 patients with bulk RNA-seq data. The p-value was calculated using the Wilcoxon signed-rank-test.

Reviewer #2

John and Poos et al uncover features of spatial heterogeneity in the subclonal architecture and molecular signatures in both tumor and microenvironment using paired specimens from 15 newly diagnosed MM patients. They show that spatial tumor heterogeneity is a frequent event in MM patients with focal lesions. Using a comprehensive approach, they discovered key features in spatial heterogeneity including down-regulation of chemoattractant cytokines and depletion of myeloid lineage cells in focal lesions. These findings can provide some insights in the processes underlying increasing independence of MM cells from the BM microenvironment. The studies are very interesting and novel but preliminary given the number of samples. A few comments are included below:

1) *The number of samples is too small to make strong statements especially that not all samples had all the multiomic analyses described.*

Response: We thank the reviewer for the overall positive feedback and the very helpful comments! We appreciate the reviewer's comment that the results need to be interpreted with

caution due to the limited sample size. Accordingly, we have re-phrased and tuned down our statements in the manuscript, e.g. now we say “we frequently observed spatial tumor heterogeneity in MM patients with focal lesions” (page 4, line 113) instead of “spatial tumor heterogeneity is a frequent event in MM patients.”, and “Our findings on the depletion of macrophages in the TME provide evidence for this hypothesis” (page 12, lines 363-364) instead of “With our second key finding we provide strong evidence”. We have also added the statements: “We appreciate that further research is required to conclude if the depletion of macrophages is the cause or the result of the expansion of advanced subclones in focal lesions and if they have any impact on therapy response and tumor cell expression profiles. The same holds true for changes in tumor cell expression profiles, including downregulation of genes coding for chemokines such as *CXCL12*.” (page 12, lines 374-378) & “Thus, we propose a spatial longitudinal study, starting at premalignant stages of the disease, with samples from different intra- and extramedullary sites to better understand the complex evolutionary processes that result in BM independence and more aggressive disease.” (page 12, lines 389-392).

On the other hand, however, we have added new flow cytometry and immunohistochemistry data to further validate our findings. Please see our response to your comment #7 and comment #2 of reviewer #3.

2) *The authors need to differentiate NA and WT in mutation calling in single-cell. Detection power should help in differentiating NA and WT. Is the fraction mutant positive expressed over WT or over WT+NA?*

Response: We thank the reviewer for this suggestion. With the aim to validate the enrichment/site-unique presence of *RAS* mutations in paired samples, which we observed using whole genome sequencing data, we used Vartrix to call mutations in the scRNA-seq data. Vartrix discriminates between NA, WT and mutation. However, since cells with a WT call presumably contained undetected mutations in *KRAS* or *NRAS*, we combined NA and WT in the first version of the manuscript. To take the reviewer’s comment into account, we now differentiate between NA and WT and describe the fraction mutant positive over cells with a WT call to show the enrichment of the respective mutations at the FL or RBM site (please see the revised Fig. 1e and the new Extended Data Fig. 2d-j). We also point out to the limitations of this approach, including the issue that due to the sparse nature of scRNA-seq even cells with a WT call could be positive for the respective mutation: “Yet, we need to note that due to the sparse nature of scRNA-seq measurements MM cells with a wildtype call presumably contained undetected *RAS*-mutations.” (page 4, lines 103-104).

Fig. 1 (upper panel) and Extended Data Fig. 2 (lower panel). In Fig. 1 we now differentiate between the respective mutation, WT and NA calls. In the Extended Data Figure 2d cells with *NRAS* mutations, WT (for *NRAS*), or NA calls are shown for patient P02 (left). In e (*KRAS*) and f (*NRAS*) the expression levels are shown for single cells of this patient with RAS mutations, WT or NA calls.

3) Related to Figure 2, are there any shared transcriptional differences (signature) observed between site-unique subclones and other disseminated clones? If any, this could provide some insights into “myeloma metastasis” and/or focal formation.

Response: In our opinion the reviewer raises a very important question. Both, site-unique and disseminated subclones, were seen in three patients of our scRNAseq set (P01, P02 and P04). In P01 the two site unique subclones showed an upregulation of *CD74* and a downregulation of ribosomal protein and HLA genes compared to the two disseminated subclones. Yet, these differences between subclones were not seen in the other two patients and we did not find a transcriptional signature across patients which could discriminate between site-unique and disseminated subclones. Unfortunately, our sample set is probably too small to adequately address this question. Therefore, we have added this statement to the Discussion: “While we identified consistent changes in expression profiles between focal lesions and paired RBM, with our limited sample set we could not address the question if there are transcriptional signatures which discriminate between site-unique and disseminated tumor subclones to gain deeper insights into “myeloma metastasis” and/or focal lesion formation. Thus, we propose a spatial-longitudinal study, starting at premalignant stages of the disease, with samples from different intra- and extramedullary sites to better understand the complex evolutionary processes that result in BM independence and more aggressive disease.” (page 12, lines 386-392).

4. It’s difficult to understand the story on interferon and HLA. It seems that the difference in transcriptional signature observed between these two CNA-defined subclones could reflect the transcriptional plasticity induced by the microenvironment. Did you see any epigenomic mechanism suggested by your scATAC-seq data for these subclones?

Response: We thank the reviewer for this valuable suggestion! We have added additional analyses according to the reviewer’s suggestions and have substantially revised this paragraph. Specifically, we have performed a combined scRNA- and scATAC-seq analysis of genetically identical subclones at different bone marrow sites to address the mechanism underlying differential expression. Using a threshold of 50 cells per subclone, this combined analysis was possible for one patient with and one patient without pronounced spatial

heterogeneity between bone marrow sites. For the patient with very similar scRNA-seq profiles (P05) genetically identical subclones from different bone marrow sites were also assigned to the same chromatin accessibility cluster. This is now shown in Extended Data Fig. 7b.

Extended Figure 7b+d. Patterns of spatial heterogeneity at the epigenetic level. (b) Single-cell ATAC-seq data for patient P05 is shown as an example for very similar chromatin accessibility profiles of genetically identical subclones at different bone marrow sites. *Left panel:* chromatin accessibility clusters and copy number aberration (CNA)-defined subclones. *Right panel:* scATAC-seq heatmap of differential accessibility peaks across subclones. Color indicates the column Z-score of normalized peak accessibility. In (d) the same plots as in (b) are shown for patient P03, who presented with pronounced epigenetic and expression differences between genetically identical subclones at different bone marrow sites. Please note that the cell number of subclone 6 in the FL was <50 cells.

In the patient with pronounced spatial heterogeneity in transcriptional and epigenetic/chromatin accessibility profiles (P03, Fig. 4b and new Extended Data Fig. 7d) differences in gene expression included an upregulation of MYC-target genes at the random bone marrow site as well as an upregulation of IFN γ and IFN α -pathway genes in the focal lesion. The predicted motif activity of transcription factors, which regulate interferon, such as IRF4, 8 and 9 was increased at the focal lesion site, suggesting epigenetic changes to underlie upregulation of the interferon pathway. Furthermore, we found CD38 to be upregulated in the focal lesion. To delineate the underlying mechanism, we assessed chromatin accessibility at the regulatory elements of CD38. We observed a strong correlation between the CD38 promoter and a distal putative enhancer as well as a regulatory intronic element only in the focal lesion. In addition, the CD38 promoter peak overlapped with IRF4 peaks in published chromatin immunoprecipitation with sequencing data for the MM cell line KMS12BM, suggesting a link between increased IRF4 activity and overexpression of CD38. We present these results in the new Figure 4c-d.

Fig. 4c-d: Paired scATAC-seq data for patient P03. In (c) a comparison between paired samples is shown for transcription factor (TF) motif deviation scores. The top TFs are marked. In (d) chromatin accessibility at the CD38 promoter plus 50000 bps upstream and downstream is depicted. The CD38 promoter peak is highlighted in light orange. *Top panel:* aggregated pseudo-bulk scATAC-seq tracks at random bone marrow (RBM, dark grey) and focal lesion (FL, light grey). *Right panel:* violin plots showing normalized CD38 expression from scRNA-seq data per spatial site. *Middle panel:* peaks are colored based on the location of the peak in either promoter (orange), distal (red), exonic (blue) or intronic (green) regions. IRF4 ChIP-seq peaks from the multiple myeloma cell line KMS12BM are shown in purple; *Bottom panel:* peak co-accessibility in the CD38 region at both sites colored by Pearson correlation coefficient.

5. It would be great to examine the protein expression level of CXCR7/CXCL12. DESeq2 should incorporate sample purity as a covariate at the very least.

Response: We thank the reviewer for this comment! First, we estimated the sample purity using histograms for the variant allele frequency of somatic mutations. These values are shown in Suppl. Table 12. We extended the data set of bulk RNA-seq by including paired samples from two more patients with paramedullary disease (please also see our response to critique #3 of reviewer #1). Next, we used purity values as a covariate for the DESeq2 analysis. While the number of differentially expressed genes decreased to 47, the top genes CXCL7, CXCL12, MYLIP and ADM were still significant.

We have used immunohistochemistry (IHC) and enzyme-linked immunosorbent assays (ELISAs) to examine the protein expression of CXCL7 and CXCL12. Unfortunately, myeloma cells were negative in histology sections stained for CXCL7 (please see an example in the response to comment #5 of reviewer #1). Furthermore, CXCL7 was at or even below the detection limit of the ELISA kit in a test set including $1-2 \times 10^5$ CD138-sorted myeloma cells from four newly diagnosed patients and bone marrow serum samples were not available. Hence, unfortunately we could not validate CXCL7 expression differences at the protein level. For CXCL12, we surprisingly did not find a significant down-regulation in sorted MM cells from focal lesions using ELISAs ($n=6$ patients, $p=0.46$ in linear mixed-effects model). This was in line with findings from stained histology sections which did not show consistent differences in the proportion of CXCL12-positive MM cells (focal lesion median: 5 (0.04-20.6) vs. random bone marrow median: 1.35 (0.04-20.1), $p=0.64$ in Wilcoxon signed rank test). The new results are shown in Fig. 3d and Extended Data Fig. 5.

Extended Data Fig. 5b (left) and Fig. 3d (right). In the extended data figure, the ratios for CXCL12 protein expression in CD138-enriched myeloma cells in paired random bone marrow (RBM) and focal lesion (FL) samples from 6 patients are shown. Expression was quantified in two experiments with duplicates using ELISA. The dotted line marks a ratio of 1, corresponding to no differences between paired samples. The p-value was calculated using a linear mixed-effects model. In **Fig. 3d** the proportion of CXCL12-positive myeloma cells is shown. In the left panel paired tissue slides from patient P13, which were stained for CXCL12, are depicted as examples. The line plot in the right panel shows the proportion of CXCL12-positive myeloma cells in paired samples from 8 patients, which were selected based on availability of bulk RNAseq. The p-value was calculated using the Wilcoxon signed-rank-test.

To address the mechanism underlying the difference between mRNA and protein levels of CXCL12, we first analysed the expression of the CXCL12 receptors CXCR4 and CXCR7. As shown in more detail in the response to your comment #6, we did not see significant differences between paired samples for CXCR4 expression at the mRNA and protein level. Interestingly, using RNA-seq we found a 5-fold upregulation of the CXCL12 scavenger-receptor CXCR7 in the lesion of patient P13, which presented with a 215-fold down-regulation of the *CXCL12* gene, indicating increased uptake of CXCL12 at this site. However, this pattern was seen in only one patient.

Next, we performed flow cytometry to quantify the proportions of mesenchymal stromal cells (MSCs) (CD271+, CD90+), since they are a major source of CXCL12 in the bone marrow¹. These cells were enriched in focal lesions in 8 patients with paired samples (mean random bone marrow=0.08% (range: 0.004-0.23%) vs mean focal lesion=2.50% (0.06-9.12%), $p=0.04$ in two-sided Wilcoxon signed rank test), indicating increased supply with CXCL12 at these sites. Please see the new Extended Data Fig. 5e for results.

Extended Data Fig. 5e: Proportion of mesenchymal stromal cells (MSCs) in the tumor microenvironment. MSCs were quantified in paired samples from 8 patients using flow cytometry. The p-value was determined using a two-sided Wilcoxon signed rank test.

We appreciate that further experiments are required to fully understand the difference between mRNA and protein levels of CXCL12 and to conclusively show the link between the presence of MSCs and intracellular CXCL12 levels of myeloma cells. Therefore, we removed downregulation of chemokine expression in focal lesions as a key finding from the Abstract and the Discussion and added the following sentences to the Discussion: “We could not validate this finding at the protein level but our preliminary findings on changes in myeloma cells and the TME provide potential explanations for this discrepancy. De Jong *et al.* recently demonstrated the important role of CXCL12-producing inflammatory mesenchymal stromal cells in the pathogenesis of MM²⁷. Here, we show a significant increase in the proportion of mesenchymal stromal cells in the TME of focal lesions and upregulation of the CXCL12-scavenger receptor CXCR7 in the patient with the strongest down-regulation of CXCL12 at the mRNA level, indicating increased supply with CXCL12 and/or uptake of the chemokine by tumor cells in focal lesions. We appreciate that this finding needs to be confirmed in larger studies.” (page 12, lines 378-385).

6. *CXCR4* in myeloma cells and their spatial distribution within the tissue in focal lesion compared to random bone marrow also needs to be confirmed at protein level. *CXCR4* is internalized in the marrow and RNA levels may not reflect surface protein expression.

Response: We would like to thank the reviewer for this important comment! Unfortunately, biomaterial was limited and only one aliquot with $1-2 \times 10^5$ cells for paired samples from 6 patients was available, which we used for quantification of CXCL12 protein levels. CXCR4 quantification would have required a different lysis buffer. Furthermore, in the extended bulk RNA-seq analysis, which included samples from two more samples and sample purity as a covariate, we did not see a clear difference for *CXCR4* expression between focal lesion and random bone marrow sites anymore. In line, we did not find a consistent downregulation in focal lesions using IHC (shown in the new Extended Data Fig. 5c-d). Therefore, we removed CXCR4 from the list of differentially expressed genes but present the data in the context of CXCL12 internalization.

Extended Data Fig. 5c-d. CXCR4 expression levels. CXCR4 expression in myeloma cells according to bulk RNA-seq (c) and proportion of CXCR4-positive myeloma cells in IHC (d) in paired random bone marrow (RBM) and focal lesions (FL). P-values were calculated using two-sided Wilcoxon signed rank tests.

7. Related to Figure 5b, this observation of decrease in monocyte/macrophage lineage cells was compatible with any other data derived from other experimental approaches like flow cytometry or cell counting from a smear sample?

Response: To validate our findings with a different experimental approach, we analyzed flow cytometry data for 8 patients with paired samples. We observed a trend towards a lower relative proportion of monocytes/macrophages in CD138-negative fractions of focal lesions (mean focal lesion=8.01% (range: 2.35-20%) vs. mean RBM=11.36% (6.58-25.33%), $p=0.078$ in two-sided Wilcoxon signed rank test. Notably, the two patients in this set, who did not show a depletion of macrophages, had almost the same plasma cell infiltration in the focal lesion (P11 and P18 in the figure), supporting the link between increased tumor load and relative depletion of macrophages. The results are shown in the new Extended Figure 8a-d.

Extended Figure 8a-e. Validation of scRNA-seq using flow cytometry. Boxplots for the proportion of cells in paired RBM/FL samples from 8 patients. Proportions for macrophages (a), CD4-positive T-cells (c) and CD8-positive T-cells (d) were calculated based on the fraction of CD138-negative cells,

while the proportion of plasma cells in (b) was determined including all cells. The boxplots show the median and the interquartile range, while the upper and lower whiskers show the highest and lowest value (excluding outliers), respectively. P-values were calculated using two-sided Wilcoxon signed rank tests.

8. *We need the details of linear regression to explain variation in macrophages L215–218. If the predictors are correlated, what is the variance inflation factor? What is the expected macrophage diminution factor given plasma cell increase vs observed in IHC?*

Response: We would like to apologize for the incomplete description of the model. For the revised manuscript we have calculated linear mixed-effects models, since we have analyzed at least two biopsies per patient and multiple regions for each biopsy. The linear mixed-effects models for cell proportions included the type of bone marrow sample (random bone marrow vs. focal lesion) and tumor load at investigated regions as fixed effects and patient ID as random effect. There was a (weak) correlation between tumor load and sample type (0.285, 0.201, and 0.192 for the macrophage-, CD4-, and CD8-model, respectively). However, the variance inflation factor for the two fixed effects of each model (macrophages, CD4- and CD8-positive T-cells) was <1.1. According to the model there was a strong decline in macrophages from regions with a low MM cell infiltration (<25%, interval 1) compared to sites with a high infiltration (>75%, interval 4). The respective estimated proportions were 41.04% (standard error (SE) 2.25) in interval 1 and a drop of -40.25% (SE: 1.92) in interval 4. Details of the three models are now shown in the new Suppl. Table 6.

9. *Do you have any serial samples from these patients, with progression from MGUS or SMM to the current NDMM or even after treatment, this would be really interesting?*

Response: We fully agree with the reviewer that spatial-longitudinal data for MM in its TME would be really interesting. Unfortunately, we do not have samples from precursor stages for these patients. Only four patients of the cohort with molecular data at baseline have relapsed (two of them with scRNA-seq at baseline) so far but only random bone marrow samples with limited numbers of myeloma cells are available. We have performed WGS with these samples and observed stable evolution, selection of a minor subclone, differential response, and single-cell expansions in one patient each. As these patterns were previously described in larger cohorts by Keats *et al.*² and Rasche *et al.*³, and since we cannot present any novel observations, we decided not to include this data in the revised manuscript. We also decided to collect larger sample sets before performing spatial-longitudinal analyses of the microenvironment. Therefore, we propose in the Discussion of the revised manuscript “**Thus, we propose a spatial-longitudinal study, starting at premalignant stages of the disease, with samples from different intra- and extramedullary sites to better understand the complex evolutionary processes that result in BM independence and more aggressive disease.**” (page 12, lines 389-392).

10. *In general the manuscript is missing confidence intervals of estimates (CCF, TCR clonality, proportion of macrophages, etc) and to Cis overlap for example between BM and FL). Also distributions should be described with IQR or standard deviation or 95% central interval in addition to mean/median.*

We thank the reviewer for this important point. We now describe distributions with the range or standard deviation/error, when appropriate. Furthermore, we calculated 95% confidence

intervals (95% CI) for cancer clonal fractions of each mutation by using a recently published approach⁴. We used the intervals to classify mutations: we defined mutations as major, if the upper band of the 95% CI was ≥ 1 , and minor otherwise, as recently described by McGranahan et al.⁴. Furthermore, major mutations with a 3-fold enrichment between the paired samples were classified as enriched. Figure 1c+d as well as Extended Data Fig. 1 were updated accordingly. In addition, in Figure 5b we now used boxplots instead of barplots for cell proportions in focal lesion and random iliac crest samples, and confidence intervals were added to Suppl. Table 5.

11. *Check the clone names used in IHC studies. The clone names for antibodies of CD4, CD8 and CD68 are the same one 'MIB-1'.*

Response: We thank the reviewer for pointing to this issue. We have changed the clone names in the methods section accordingly.

12. *Related to Supple Table 7, there is no T cell subtype detail yet the legend mentions it.*

Response: We thank the reviewer for spotting this. The T cell subtype information for each expanded TCR clone is now provided in Suppl. Table 8.

Reviewer #3

In this study, John et al. attempted to decipher spatial heterogeneity of multiple myeloma (MM) and its microenvironment between random bone marrow (RBM) samples from the iliac crest and paired focal lesion (FL) specimens at single-cell level. The manuscript is well written and documented with informative figures. This study generated a large amount of data, and the huge amount efforts spent for this work need to be recognized. While the experimental and bioinformatical effort invested into this paper is remarkable, its results are not truly novel as many of them have been described in previous studies (to which the authors also refer to). Overall, the authors attempted to elucidate a detailed description on subclonal architecture, expression signatures and the composition of the microenvironment in FL, which are quite descriptive. Furthermore, the major conclusions in this study are drawn from a small sample size (only 5 or 6 patients for single-cell sequencing) and lack further validation. Many claims are speculative and are not tested, nor based on independent cohorts neither experimentally. Many further studies are required to validate the data and understand the interactions within immune cell populations (e.g., the depleted macrophages and skewed T-cell repertoire in FL) and between the host immune system and the tumor. Furthermore, there is little novel insight from the study, and the low number of patients undermined all conclusions. This paper is more suited for a more specialized journal.

Response to general comment: First, we would like to thank the reviewer for the constructive criticism and suggestions for additional analyses, which, in our opinion, helped to significantly improve our manuscript. We agree with the reviewer that the description of spatial genomic heterogeneity in myeloma, as presented in the first paragraph, is not novel. Yet, we think that our observations are still important, as we 1) confirm for the first time the retrospectively collected data by Rasche *et al.* with samples from a prospective study, 2) modify and in part even disprove the conclusion drawn by Merz *et al.* that spatial heterogeneity is absent in

intramedullary lesions and exclusively seen in extramedullary disease, and 3) show that pronounced spatial genomic heterogeneity is not necessarily associated with site-specific T cell clones as seen in solid tumors.

Although tumor sequencing results strongly suggest that focal lesions are hotspots of tumor evolution, they are still a poorly understood imaging phenomenon. Radiology data suggest that they are areas with increased cellularity and metabolic activity. However, the cellular composition of focal lesions was unknown when we conducted our study. While a single-cell analysis using paired samples from different bone marrow sites has recently been published by Merz *et al.*⁵ the authors did not find spatial genomic heterogeneity, probably due to the absence of (large) focal lesions at the investigated sites. Therefore, we are the first group to publish single cell data on spatially heterogeneous myeloma samples. Thus, to the best of our knowledge most results in our manuscript have not been described in previous studies, including gene expression changes in focal lesions, the composition of the TME in focal lesions and the depletion of macrophages at sites with a high plasma cell infiltration as well as the number of tumor subclones at different sites in the bone marrow.

We also agree that our single-cell sequencing and bulk RNAseq set is based on a limited sample size. However, focal lesions are often accessible by biopsy only with an immanent medical risk to the patient that is not acceptable for a purely diagnostic study, making this type of study highly challenging. Furthermore, single-cell sequencing requires high-quality cellular analytes, making our paired sample set even more precious.

Despite the limited sample size, we identified changes in tumor gene expression profiles and the TME that could be confirmed using larger and/or independent sample/data sets. This includes a depletion of macrophages in the TME, which we now validate using flow cytometry and immunohistochemistry for paired samples from 8 and 21 patients, respectively. Gene expression differences between focal lesion and random bone marrow samples were validated using paired samples from 250 patients. Although we could not confirm downregulation of chemokines in focal lesion myeloma cells at the protein level, we provide new data that could potentially explain this difference between mRNA and protein (please see our response to your comment #4).

Last but not least we agree with the reviewer that many further studies are required to elucidate the mechanisms underlying the evolution of myeloma cells in focal lesions and to understand the interactions of myeloma cells with immune cell populations. Here, we present the subclonal architecture and the composition of the TME at different sites in the bone marrow as a first (important) step to improve our understanding of focal lesions, which are associated with poor outcome in myeloma. As one possible next step, “Thus, we propose a spatial-longitudinal study, starting at premalignant stages of the disease, with samples from different intra- and extramedullary sites to better understand the complex evolutionary processes that result in BM independence and more aggressive disease.” in the Discussion (page 12, lines 389-392). Please also see our responses to your specific comments for a more detailed discussion of new results and limitations.

Specific comments:

1. *The authors claims that the genetically identical subclones at different BM sites have different gene expression signatures, which is contributed by transcriptional plasticity. This conclusion is one of the main findings of this study, but it is very obvious indeed. Of course, spatially different TME has different gene expression signatures. What is the underlying molecular mechanism? The authors did not elaborate it adequately.*

We would like to thank the reviewer for the opportunity to clarify this point. Using bulk RNA-seq we observed differences in expression profiles between paired focal lesion/random bone marrow samples, yet it was not clear, if these differences were just due to the presence of unique subclones at these sites or the result of transcriptional or epigenetic plasticity of myeloma cells. To address this question, we compared genetically identical subclones in paired samples. Of note, we observed two patterns, including very similar profiles and pronounced spatial heterogeneity in expression profiles. For the revised manuscript, we have analyzed corresponding scATAC-seq data and made the same observation, which is shown in the new Extended Data Fig. 7b+d.

Extended Figure 7b+d. Patterns of spatial heterogeneity at the epigenetic level. (b) Single-cell ATAC-seq data for patient P05 is shown as an example for very similar chromatin accessibility profiles of genetically identical subclones at different bone marrow sites. *Left panel:* chromatin accessibility clusters and copy number aberration (CNA)-defined subclones. *Right panel:* scATAC-seq heatmap of differential accessibility peaks across subclones. Color indicates the column Z-score of normalized peak accessibility. In (d) the same plots as in (b) are shown for patient P03 as an example for pronounced differences in epigenetic and expression profiles between genetically identical subclones at different bone marrow sites. Please note that the cell number of subclone 6 in the FL was <50 cells.

Thus, despite differences in the composition of the TME, genetically identical subclones can show the same/similar expression and epigenetic profiles at different bone marrow sites, a pattern which was seen in two patients. In contrast, pronounced spatial heterogeneity in epigenetic and transcriptional profiles of identical subclones at different BM sites was seen in two other patients. To the best of our knowledge there is no report in MM which conclusively demonstrates that the same subclone can show transcriptional heterogeneity at different sites. This finding may seem trivial, however the differential expression of immunotherapy targets such as CD38 and BCMA has wide ranging implications for the treatment of myeloma patients,

who are regularly treated with corresponding monoclonal and bispecific antibodies or CAR-T-cells against these targets.

To take the reviewer's comment into account that the underlying mechanisms need to be addressed, we have performed a combined analysis of scRNA-seq and scATAC-seq in a patient with pronounced spatial heterogeneity in expression profiles. Differences in gene expression included an upregulation of MYC-target genes at the RBM-site as well as an upregulation of IFN γ and IFN α -pathway genes in the focal lesion. The predicted motif activity of transcription factors, which regulate interferon, such as IRF4, 8 and 9, was increased at the focal lesion site, suggesting epigenetic changes to underlie upregulation of the interferon pathway. These results are shown in the new Fig. 4c. Furthermore, we found the immunotherapy target *CD38* among the top differentially expressed genes between genetically identical subclones at different bone marrow sites. To further delineate the mechanism underlying increased *CD38* expression at the focal lesion site, we assessed chromatin accessibility at the regulatory elements of *CD38*. We observed a strong correlation between the *CD38* promoter and a distal putative enhancer as well as a regulatory intronic element only in the focal lesion. In addition, the *CD38* promoter peak overlapped with IRF4 peaks in published chromatin immunoprecipitation with sequencing (ChIP-seq) data for the MM cell line KMS12BM⁶, suggesting a link between increased IRF4 activity and overexpression of *CD38*. These new observations are depicted in Fig. 4d.

Fig. 4c-d: Paired scATAC-seq data for patient P03. In (c) a comparison between paired samples is shown for transcription factor (TF) motif deviation scores. The top TFs in the focal lesion are marked. In (d) chromatin accessibility at the *CD38* promoter plus 50000 bps upstream and downstream is depicted. The *CD38* promoter peak is highlighted in light orange. *Top panel:* aggregated pseudo-bulk scATAC-seq tracks at random bone marrow (RBM, dark grey) and focal lesion (FL, light grey). *Right panel:* violin plots showing normalized *CD38* expression from scRNA-seq data per spatial site. *Middle panel:* peaks are colored based on the location of the peak in either promoter (orange), distal (red), exonic (blue) or intronic (green) regions. IRF4 ChIP-seq peaks from the multiple myeloma cell line KMS12BM are shown in purple; *Bottom panel:* peak co-accessibility in the *CD38* region at both sites colored by Pearson correlation coefficient.

To elucidate if changes in cell-cell interaction profiles could explain spatial differences in expression profiles, we performed cell interaction analyses using CellChat and paired scRNA-seq data. We found differences between paired samples of single patients (e.g. interactions mediated by *ICAM1*), but there were no MM-TME interactions that were characteristic for focal lesions and random bone marrow, respectively, suggesting that differential expression of genes such as *CXCL7* or *CXCL12* is not due to specific myeloma-TME interactions. However, we fully appreciate the limitations of this analysis and decided not to present these new results in the revised manuscript. Instead, we propose a larger spatial-longitudinal study, which would

potentially allow us to analyse changes in these interactions over time and space and the impact on tumor gene expression.

Example figures. Predicted interactions between myeloma and TME cells. In patient P01 (upper figure) ICAM1-mediated interactions were predicted only for the focal lesion (right panel). For patient P02 (lower figure) the opposite pattern was predicted.

2. To explain why macrophages are depleted in focal lesions, the authors correlated the proportion of macrophages with the level of plasma cell infiltration. However, this result seems not convincing. In fig5c, there is almost no extra space for other cells when the infiltration of

plasma cells is greater than 76%, thus the decreased proportion of macrophages or other cells is as expected/not surprising. Besides, in Suppl. Table 8, the proportions of plasma cells in RBM and FL pair do not show great difference. Could the authors provide the mechanism explanation why macrophages are depleted in intramedullary focal lesions?

Response: We would like to apologize that this validation step was not clearly presented in the first version of the manuscript. Initially, we aimed to validate our finding that macrophages are depleted in the tumor microenvironment of focal lesions. Therefore, we stained CD68, CD4, and CD8 in core biopsies from the 6 patients with scRNA-seq to quantify macrophages as well as CD4- and CD8-positive T-cells as control cells, respectively. In a preliminary analysis we found regions with low numbers of macrophages in both random bone marrow biopsies and focal lesions. Since these regions were characterized by a high level of plasma cell infiltration and focal lesions typically show a high tumor load, we addressed the question if low numbers of macrophages were indeed a unique feature of focal lesions or just associated with the level of plasma cell infiltration. Therefore, we quantified the cells in regions with different levels of plasma cell infiltration, which we defined in four intervals (1-25%, 26-50%, 51-75%, and 76-100%). We agree that there is almost no extra space for other cells when the infiltration of plasma cells is greater than 75%. However, calculating models for macrophages and T-cells, we observed a stronger decline of macrophages from regions with a low myeloma cell infiltration (<25%, interval 1) to regions with a high infiltration (>75%, interval 4) compared to CD4- and CD8-positive T-cells in both focal lesion and random bone marrow sites. And we show that even in regions with the highest level of plasma cell infiltration (interval 4) all three cell types are usually still detectable. We concluded that the stronger decline of macrophages could explain the relative depletion in our initial scRNA-seq analysis of the tumor microenvironment.

In the revised version of the manuscript we have extended the immunohistochemistry analysis to 21 patients with paired samples, which confirmed the stronger depletion of macrophages in regions with a high plasma cell infiltration compared to CD8-positive T-cells. This is now shown in Fig. 5d. Furthermore, we have quantified the myeloma cells in these patients, which shows that focal lesions have a significantly increased tumor load (please see the new Fig. 5e). Suppl. Table 5 shows the proportion of cells in CD138-depleted cell fractions in paired samples. To avoid confusion, we have marked plasmablasts and defined them as normal precursors of plasma cells in the revised manuscript.

Fig. 5. (d-e) Boxplots for the proportion of macrophages, CD4- and CD8-positive T-cells according to IHC for 21 patients with paired RBM/FL samples. Regions in stained slides were classified according to the level of plasma cell infiltration, which was scored from 1 to 4, representing 5-25%, 26-50%, 51-75% and 76-100%, respectively. Usually, multiple regions with a different level of plasma cell infiltration were considered for each slide. **(e)** Boxplot for the total plasma cell infiltration in paired slides from the same 21 patients. The boxplots show the median and the interquartile range, while the upper and lower whiskers show the highest and lowest value (excluding outliers), respectively.

Furthermore, we used flow cytometry to quantify monocytes/macrophages and T-cells. Including paired samples from 8 patients, there was a strong trend ($p=0.08$) to a lower proportion of monocytes/macrophages in the CD138-depleted cell fraction of focal lesions. Of note, the two pairs without differences in the proportion of monocytes/macrophages had almost the same myeloma cell infiltration in the paired samples, further supporting a link between tumor load and relative proportions of monocytes/macrophages. These results are now depicted in Extended Data Fig. 8a+c.

Extended Data Fig. 8. Cell proportions in paired samples according to flow cytometry. (a-d) Boxplots for the proportion of cells in paired RBM/FL samples from 8 patients. Proportions for macrophages **(a)**, CD4-positive T-cells **(c)** and CD8-positive T-cells **(d)** were calculated based on the fraction of CD138-negative cells, while the proportion of plasma cells in **(b)** was determined including all cells. The boxplots show the median and the interquartile range, while the upper and lower whiskers show the highest and lowest value (excluding outliers), respectively. P-values were calculated using two-sided Wilcoxon signed rank tests.

Our approach represents a snapshot of ongoing evolution. Therefore, it is unfortunately difficult or even impossible to address the mechanism underlying the depletion of macrophages in focal lesions with the available data. We can only speculate that macrophages are preferentially replaced by invading or expanding tumor cells. Alternatively, there could be an enrichment of other cell types in focal lesions, as shown in our preliminary analysis for mesenchymal stromal cells (please see the new Extended Data Fig. 5e).

Extended Data Fig. 5e: Proportion of mesenchymal stromal cells (MSCs) in the tumor microenvironment. MSCs were quantified in paired samples from 8 patients using flow cytometry. The p-value was determined using a two-sided Wilcoxon signed rank test.

Since we cannot conclude if changes in the tumor microenvironment and tumor cell expression profiles are the cause or the result of the expansion of advanced subclones in focal lesions, we have edited the Discussion and “Thus, we propose a spatial-longitudinal study, starting at premalignant stages of the disease, with samples from different intra- and extramedullary sites to better understand the complex evolutionary processes that result in BM independence and more aggressive disease.” (page 12, lines 389-392).

3. *The analysis of scATAC-seq data is superficial. The data were used only for CNA-calls. Is there other useful information that could be drawn from scATAC-seq? For example, the integration of the RNA and ATAC-seq data would benefit the molecular mechanism analysis.*

Response: We thank the reviewer for the suggestion to better integrate scRNA- and scATAC-seq data! First, we analyzed the 6 genes, which we identified as differentially expressed between focal lesion and random bone marrow. Unfortunately, in scATAC-seq there was low coverage at the *CEBPD*, *CXCL7* and *CXCL12* gene loci, while the other three genes showed no differential accessibility in regulatory regions between focal lesions and RBM (Extended Data Fig. 6b). Next, we extended the comparison of genetically identical subclones at different bone marrow sites by including scATAC-seq data. In line with scRNA-seq, we found two patterns, including very similar profiles and pronounced spatial heterogeneity in accessibility profiles, with the latter suggesting changes induced by the tumor microenvironment. Please see the new Extended Data Fig. 7 b+d.

Extended Data Figure 7b+d. Patterns of spatial heterogeneity at the epigenetic level. (b) Single-cell ATAC-seq data for patient P05 is shown as an example for very similar chromatin accessibility profiles of genetically identical subclones at different bone marrow sites. *Left panel:* chromatin accessibility clusters and copy number aberration (CNA)-defined subclones. *Right panel:* scATAC-seq heatmap of differential accessibility peaks across subclones. Color indicates the column Z-score of normalized peak accessibility. In (d) the same plots as in (b) are shown for patient P03 as an example for pronounced differences in epigenetic and expression profiles between genetically identical subclones at different bone marrow sites. Please note that the cell number of subclone 6 in the FL was <50 cells.

Differences in gene expression between genetically identical subclones at different bone marrow sites in patient P03 included an upregulation of IFN γ and IFN α -pathway genes as well as CD38 in the focal lesion. Using scATAC-seq, we observed increased motif activity of transcription factors, which regulate interferon, such as IRF4, 8 and 9, suggesting epigenetic changes to underlie upregulation of the interferon pathway. To delineate the mechanism underlying increased CD38 expression at the focal lesion site, we assessed chromatin accessibility at the regulatory elements of CD38. We observed a strong correlation between the CD38 promoter and a distal putative enhancer as well as a regulatory intronic element only in the focal lesion. In addition, the CD38 promoter peak overlapped with IRF4 peaks in published chromatin immunoprecipitation with sequencing (ChIP-seq) data for the MM cell line KMS12BM, suggesting a link between increased IRF4 activity and overexpression of CD38. The results are now presented in Fig. 4c-d.

Fig. 4: Paired scATAC-seq data for patient P03. In (c) a comparison between paired samples is shown for transcription factor (TF) motif deviation scores. The top TFs in the focal lesion are marked. In (d) chromatin accessibility at the CD38 promoter plus 50000 bps upstream and downstream is depicted. The CD38 promoter peak is highlighted in light orange. *Top panel:* aggregated pseudo-bulk scATAC-seq tracks at random bone marrow (RBM, dark grey) and focal lesion (FL, light grey). *Right panel:* violin plots showing normalized CD38 expression from scRNA-seq data per spatial site. *Middle panel:* peaks are colored based on the location of the peak in either promoter (orange), distal (red), exonic (blue) or intronic (green) regions. IRF4 ChIP-seq peaks from the multiple myeloma cell line KMS12BM are shown in purple; *Bottom panel:* peak co-accessibility in the CD38 region at both sites colored by Pearson correlation coefficient.

4. One of the main findings that chemoattractant cytokines CXCL7 and CXCL12 are downregulated in focal lesions, should be validated on the protein level.

Response: To validate the downregulation of the two chemokines at the protein level, we have applied ELISA and immunohistochemistry (IHC). Unfortunately, the protein levels of CXCL7 in $1-2 \times 10^5$ CD138-enriched bone marrow plasma cells from four newly diagnosed myeloma patients were at or even below the detection limit of the ELISA. In stained histology sections myeloma cells were negative for CXCL7 (please see response to point #5 of reviewer #1 for an example). Thus, and also due to limited patient material, we excluded CXCL7 from the validation step.

For CXCL12, expression levels were in general also quite low, however it was detectable with both techniques. We surprisingly did not find a significant difference at the protein level between paired samples using ELISA and IHC as shown in the new Fig. 3d and Extended Data Fig. 5.

Extended Data Fig. 5 (left) and Fig. 3d (right). In the extended data figure, the ratios for CXCL12 protein expression in CD138-enriched myeloma cells in paired random bone marrow (RBM) and focal lesion (FL) samples from 6 patients are shown. Expression was quantified in two experiments with duplicates using ELISA. The dotted line marks a ratio of 1, corresponding to no differences between paired samples. The p value was calculated using a linear mixed-effects model. In **Fig. 3d** the proportion of CXCL12-positive myeloma is shown. In the left panel paired tissue slides from patient P13, which were stained for CXCL12, are depicted as examples. The line plot in the right panel shows the proportion of CXCL12-positive myeloma cells in paired samples from 8 patients, which were selected based on availability of bulk RNAseq. The p value was calculated using the Wilcoxon signed-rank-test.

To address the mechanism underlying the difference between mRNA and protein level, we first analyzed the expression level of the CXCL12 receptors CXCR4 and CXCR7. For CXCR4, we found up to 2-fold differences between paired samples from 11 patients using bulk RNAseq but the changes were seen in both directions (7x down- and 4x upregulated in focal lesions). A similar pattern was seen for the proportion of CXCR4-positive MM cells in IHC.

CXCR4 expression levels: CXCR4 expression in myeloma cells according to bulk RNA-seq (left panel) and proportion of CXCR4-positive myeloma cells in IHC (right panel) in paired random bone marrow (RBM) and focal lesions (FL).

These observations do not support increased internalization via this receptor. Interestingly, the CXCL12 scavenger-receptor CXCR7 showed a 5-fold upregulation at the mRNA level in the paramedullary lesion of patient P13. The lesion presented with a 215-fold down-regulation of the *CXCL12* gene compared to the RBM but showed similar values at the protein level

according to IHC and ELISA (please see the new Extended Data Fig. 5b and Fig. 3d), indicating increased uptake in this patient. Yet, significant upregulation of CXCR7 in focal lesions was only seen in this patient.

Next, we quantified the proportion of mesenchymal stromal cells (MSCs) in paired samples from 8 patients. MSCs are an important source of CXCL12 for myeloma cells¹. Using flow cytometry, we observed a significantly increased proportion of MSCs in focal lesions (mean RBM=0.08% (range:0.004-0.23%) vs mean FL=2.50% (0.06-9.12%), p=0.04, Extended Data Fig. 5e). While this result provides one possible explanation for the difference between CXCL12 mRNA and protein levels, we appreciate that it is based on a small sample set and needs to be confirmed experimentally and in larger studies. Therefore, we removed downregulation of chemokine expression in focal lesions as a key finding from the Abstract and the Discussion and added the following sentences to the Discussion: “We could not validate this finding at the protein level but our preliminary findings on changes in myeloma cells and the TME provide potential explanations for this discrepancy. De Jong *et al.* recently demonstrated the important role of CXCL12-producing inflammatory mesenchymal stromal cells in the pathogenesis of MM²⁷. Here, we show a significant increase in the proportion of mesenchymal stromal cells in the TME of focal lesions and upregulation of the CXCL12-scavenger receptor CXCR7 in the patient with the strongest down-regulation of CXCL12 at the mRNA level, indicating increased supply with CXCL12 and/or uptake of the chemokine by tumor cells in focal lesions. We appreciate that this finding needs to be confirmed in larger studies.” (page 12, lines 378-385).

We would also like to note that focal lesions are often only accessible with a risk to the patient that is not acceptable for a purely diagnostic study, complicating this type of study and making our paired sample set highly precious. We also discuss this issue in the revised manuscript. (page 12, lines 392-394).

5. Line 123, the name of “clone 6B” is abrupt, which should be replaced by “subclone 6b” as the same in Fig.2C.

Response: Thank you for pointing to this typo. We have edited the manuscript accordingly.

6. Line 463, there is an incomplete expression: “multimodal reference mapping approach from”.

Response: Thank you! We have completed the expression in the revised manuscript: “multimodal reference mapping approach from Stuart and co-workers”.

7. The order of group color should keep consistent in Fig. 1c.

Response: As requested, we keep the order consistent in the revised Fig. 1c.

8. The color of Fig. 2b is too miscellaneous.

Response: We have changed the colors to black and grey.

References

1. de Jong, M. M. E. *et al.* The multiple myeloma microenvironment is defined by an inflammatory stromal cell landscape. *Nat. Immunol.* **22**, 769–780 (2021).
2. Keats, J. J. *et al.* Clonal competition with alternating dominance in multiple myeloma. *Blood* **120**, 1067–1076 (2012).
3. Rasche, L. *et al.* The spatio-temporal evolution of multiple myeloma from baseline to relapse-refractory states. *Nat. Commun.* **13**, 4517 (2022).
4. McGranahan, N. *et al.* Clonal status of actionable driver events and the timing of mutational processes in cancer evolution. *Sci. Transl. Med.* **7**, 283ra54 (2015).
5. Merz, M. *et al.* Deciphering spatial genomic heterogeneity at a single cell resolution in multiple myeloma. *Nat. Commun.* **13**, 807 (2022).
6. Jin, Y. *et al.* Active enhancer and chromatin accessibility landscapes chart the regulatory network of primary multiple myeloma. *Blood* **131**, 2138–2150 (2018).

Reviewers' Comments:

Reviewer #2:

Remarks to the Author:

The authors answered most of my concerns at the initial review by adding new experimental data. The huge amount of efforts spent for this work need to be recognized. I believe this study is well-written and thought-provoking, promoting further spatial-longitudinal studies to better understand myeloma pathophysiology. A few minor comments are included below:

#1. As for the discussion on the discrepancy between mRNA (down-regulation in focal lesions) and protein (similar) levels of CXCL12, I'm still confused. The authors provided the potential explanation that the enrichment of MSCs (CXCL12 source) in the TME of focal lesions. And they provided the evidence of enrichment of MSCs in FL. Then, I'm wondering if they can add double staining of CXCL12 and CD138 for Fig.3d like they did for Fig.3c. This way allows them to compare CXCL12 protein level in both CD138pos and CD138neg fraction. This may provide more solid evidence supporting this potential explanation.

Reviewer #3:

Remarks to the Author:

In the revised submission, the authors have made efforts to address the concerns raised in the initial review by conducting additional analyses and including a theoretical discussion on the novelty and limitations of their study. The authors have incorporated several crucial analyses that were absent in the previous version. These include integrated analysis of epigenetic plasticity (scATAC-seq) and transcription programs (scRNA-seq) to analyze identical subclones in paired samples, quantification of monocytes/macrophages and T-cells using flow cytometry for eight additional patient samples, and validation of chemoattractant cytokines CXCL7 and CXCL12 at the protein level.

The authors have also made an effort to address the molecular mechanisms underlying the transcriptional differences among subclones, addressing the concerns raised in my earlier comments. They have compared chromatin accessibility and cell-cell communications. However, perhaps due to the small sample size, they were unable to identify any clear patterns across patients. Thus, it is not clear that how can these conclusions contribute to the broad multiple myeloma field and contribute to molecular mechanism study or potential therapeutics? One possibility may be to validate those transcription programs/patterns among the subclones they identified in independent cohorts and link them to molecular or clinical phenotypes.

The authors also observed T-cell expansion within both focal lesions and RBM, with the degree of expansion varying among patients. They found that these TCR expansions were not related to genetic heterogeneity. Could this expansion be related to the generation of tumor neoantigens? The WES and bulk RNA-seq data may be used for neoantigen discovery and test for the relationship. Also, the limited sample size may account for the failure to identify the mechanisms behind this T-cell expansion.

Overall, although the revised manuscript still lacks mechanistic insights and is limited by small sample size, resulting in a primarily descriptive research article, it still presents some novelty, particularly as the first single-cell profiling work on spatially heterogeneous myeloma samples.

Reviewer #4:

Remarks to the Author:

This detailed report characterizes the heterogeneity of paired bone marrow and focal lesion samples of

newly diagnosed multiple myeloma patients. The report provides an elegant and thorough description of the authors' analyses of gene expression signatures, subclonal architecture and the marrow microenvironment in primary samples.

The initial reviews of the paper requested clarifications of methodology and comment on the impact of the work. The authors provide thorough responses to the original reviews, and have included additional supplementary data, which improves the paper. Their edits to the paper in response to these specific requests have also improved the manuscript. The revised descriptions of differential expression signatures associated within focal lesions and macrophage depletion of focal lesions are particularly improved.

While this work remains largely descriptive and does have limitations (such as lack of protein level validation of chemokine downregulation and an overall small sample size), it is hypothesis-generating and valuable. The authors have significantly improved the paper by providing critical detail to the methods section and edited the discussion to acknowledge the small sample size. They have also clarified their explanations of the potential mechanisms for the observed macrophage and T-cell repertoire changes in sites of high plasma cell burden.

Revision of manuscript # NCOMMS-22-13015A

REVIEWERS' COMMENTS

Reviewer #2

The authors answered most of my concerns at the initial review by adding new experimental data. The huge amount efforts spent for this work need to be recognized. I believe this study is well-written and though-provoking, promoting further spatial-longitudinal studies to better understand myeloma pathophysiology. A few minor comments are included below:

1) As for the discussion on the discrepancy between mRNA (down-regulation in focal lesions) and protein (similar) levels of CXCL12, I'm still confused. The authors provided the potential explanation that the enrichment of MSCs (CXCL12 source) in the TME of focal lesions. And they provided the evidence of enrichment of MSCs in FL. Then, I'm wondering if they can add double staining of CXCL12 and CD138 for Fig.3d like they did for Fig.3c. This way allows them to compare CXCL12 protein level in both CD138pos and CD138neg fraction. This may provide more solid evidence supporting this potential explanation.

Response: We would like to thank the reviewer for the positive feedback and the constructive critique! We agree that double staining for CXCL12 and a plasma cell marker would have been better to discriminate between myeloma and other cells. However, although double staining of CXCL12/MUM1 worked for our control samples, including specimens from liver, pancreas and tonsil (Rebuttal Fig. 1), it was not feasible to get reliable results for the myeloma samples. As shown in the figure, when performing consecutive staining of CXCL12 and MUM1, most myeloma cells in the specimen were negative for MUM1. Changing the order of staining also did not result in reliable results. Therefore, the pathologist Dr. Brobeil identified plasma cells based on their morphology, used a consecutive slide stained for MUM1 to confirm his classification, and quantified the proportion of CXCL12-positive plasma cells. We have added this information to Methods.

Rebuttal Fig. 1: Double staining of CXCL12 and MUM1.

Reviewer #3

In the revised submission, the authors have made efforts to address the concerns raised in the initial review by conducting additional analyses and including a theoretical discussion on the novelty and limitations of their study. The authors have incorporated several crucial analyses that were absent in

the previous version. These include integrated analysis of epigenetic plasticity (scATAC-seq) and transcription programs (scRNA-seq) to analyze identical subclones in paired samples, quantification of monocytes/macrophages and T-cells using flow cytometry for eight additional patient samples, and validation of chemoattractant cytokines CXCL7 and CXCL12 at the protein level.

1) The authors have also made an effort to address the molecular mechanisms underlying the transcriptional differences among subclones, addressing the concerns raised in my earlier comments. They have compared chromatin accessibility and cell-cell communications. However, perhaps due to the small sample size, they were unable to identify any clear patterns across patients. Thus, it is not clear that how can these conclusions contribute to the broad multiple myeloma field and contribute to molecular mechanism study or potential therapeutics? One possibility may be to validate those transcription programs/patterns among the subclones they identified in independent cohorts and link them to molecular or clinical phenotypes.

Response: We would like to thank the reviewer for appreciating our effort to improve the manuscript. We would also like to thank her/him for the excellent suggestion to further address the clinical and/or biological relevance of our findings. Therefore, we selected the genes, which were differentially expressed between paired samples, as well as the MHC II component *CD74*, which was frequently differentially expressed between distinct subclones or bone marrow sites. As a first step, we examined the correlation between the expression of these genes and survival in 653 newly diagnosed multiple myeloma patients with available gene expression profiling data, who had been treated within the GMMG HD4 and MM5 trials. We found that low (\leq median) expression of *CXCL7* and *CXCL12* was associated with inferior overall survival, and the same holds true for *CD74*. Next, we correlated expression with clinical parameters. While low expression of *CD74*, *CXCL7* and *CXCL12* was associated with increased plasma cell infiltration and higher revised International Staging System stages, there was also an association between low *CXCL12* expression and cytogenetic risk markers, including translocation t(4;14), deletion 17p and gain of 1q21. These new results are presented in Suppl. Fig. 5b-f and Suppl. Table 3.

We have added the following statements to Results:

“In order to understand the clinical implications of these differentially expressed genes, we correlated their expression with patients’ characteristics and outcome in a cohort of 653 newly diagnosed MM patients. Low expression (\leq median) of *CXCL7* (hazard ratio (HR): 1.39, 95% confidence interval: 1.26-1.52, $p=0.01$, Cox regression and log-rank test) and *CXCL12* (HR: 1.71 (1.58-1.84), $p<0.001$, Cox regression and log-rank test) was associated with inferior overall survival (Suppl. Fig. 5b-c). No significant effect was seen for *ADM* and *MYLIP* (Suppl. Fig. 5d-e). For *CXCL7* and *CXCL12*, low expression was associated with increased plasma cell (PC) infiltration levels and advanced disease according to the revised International Staging System (rISS) ($p<0.05$, t-test and chi-square test, respectively, Suppl. Table 3). Furthermore, low expression of *CXCL12* was associated with cytogenetic risk markers, including translocation t(4;14), deletion 17p and gain of 1q21 (all $p<0.05$, chi-square test, Suppl. Table 3).”

“Since differential expression of the MHC II component *CD74* was seen in multiple comparisons, we next examined the correlation between its expression and clinical parameters. In 653 newly diagnosed patients, low (\leq median) expression of *CD74* was associated with inferior OS (HR: 1.39 (1.25-1.52), $p=0.01$, Cox regression and log-rank test), as well as increased PC infiltration ($p=0.03$, t-test) and higher rISS stages ($p=0.047$, chi-square test, Suppl. Fig. 5f, Suppl. Table 3).”

2) The authors also observed T-cell expansion within both focal lesions and RBM, with the degree of expansion varying among patients. They found that these TCR expansions were not related to genetic heterogeneity. Could this expansion be related to the generation of tumor neoantigens? The WES and bulk RNA-seq data may be used for neoantigen discovery and test for the relationship. Also, the limited sample size may account for the failure to identify the mechanisms behind this T-cell expansion.

Response: We thank the reviewer for this important question! To address it, we predicted tumor neoantigens using WGS and bulk RNA-seq data. Unfortunately, we only observed trends for a link between the number of expanded T-cell clones and the number of predicted neoantigens (Rebuttal Fig. 2). Yet, as suggested by the reviewer, we fully appreciate that the failure to decipher the mechanisms

behind expanded T-cell clones could be due to the limited sample size in our study. Therefore, we don't show the data in the manuscript but have added a statement regarding limited sample size.

Rebuttal Fig. 2: Association between the number of expanded T-cell clones (x-axis) and the number of predicted neoantigens (y-axis) for all patients with scRNA-seq and WGS data (n=5). Each sample was plotted separately and colored by patient and location. The correlation was separately tested for focal lesion and random bone marrow samples using Pearson correlation.

3) Overall, although the revised manuscript still lacks mechanistic insights and is limited by small sample size, resulting in a primarily descriptive research article, it still presents some novelty, particularly as the first single-cell profiling work on spatially heterogeneous myeloma samples.

Response: Thank you!

Reviewer #4 (Remarks to the Author):

This detailed report characterizes the heterogeneity of paired bone marrow and focal lesion samples of newly diagnosed multiple myeloma patients. The report provides an elegant and thorough description of the authors' analyses of gene expression signatures, subclonal architecture and the marrow microenvironment in primary samples.

The initial reviews of the paper requested clarifications of methodology and comment on the impact of the work. The authors provide thorough responses to the original reviews, and have included additional supplementary data, which improves the paper. Their edits to the paper in response to these specific requests have also improved the manuscript. The revised descriptions of differential expression signatures associated within focal lesions and macrophage depletion of focal lesions are particularly improved.

While this work remains largely descriptive and does have limitations (such as lack of protein level validation of chemokine downregulation and an overall small sample size), it is hypothesis-generating and valuable. The authors have significantly improved the paper by providing critical detail to the methods section and edited the discussion to acknowledge the small sample size. They have also clarified their explanations of the potential mechanisms for the observed macrophage and T-cell repertoire changes in sites of high plasma cell burden.

Response: We would like to thank the reviewer for the evaluation of our paper and the overall positive feedback!